



# Retrieval of microphysical cloud parameters from EM-FTIR spectra measured in Arctic summer 2017

Philipp Richter[1], Mathias Palm[1], Christine Weinzierl[1], Hannes Griesche[2], Penny M. Rowe[3], and Justus Notholt[1]

[1]University of Bremen, Institute of Environmental Physics, Otto-Hahn-Allee 1, 28359 Bremen
[2]Leibniz Institute for Tropospheric Research (TROPOS), Permoserstr. 15, 04318 Leipzig
[3]NorthWest Research Associates, Redmond, WA, USA

**Correspondence:** Philipp Richter (phi.richter@iup.physik.uni-bremen.de)

**Abstract.** Infrared spectral radiances of optically thin clouds show high sensitivity to changes of the microphysical cloud parameters. Therefore, measurements of infrared spectral radiance of clouds in the spectral range from $770.9\,\mathrm{cm}^{-1}$ to $1163.4\,\mathrm{cm}^{-1}$ using a mobile Fourier Transform Infrared spectrometer were performed on the German research vessel *Polarstern* in the Arctic in summer 2017.

A new retrieval of microphysical cloud parameters from optically thin clouds called *Total Cloud Water retrieval*, designed to retrieve cloud water optical depth $\tau_{cw}$, total effective droplet radius $r_{total}$ and condensed water path $CWP$ from infrared spectral radiances without the incorporation of spectral radiances in the far-infrared below $600\,\mathrm{cm}^{-1}$, has been developed for application on radiances from the measurement campaign. Validation is performed against derived quantities from a combined cloud radar, lidar and microwave radiometer measurement synergy retrieval, called Cloudnet, performed by the Leibnitz Institute for Trospheric Research.

Applied to spectral radiances of synthetic testcases, *Total Cloud Water retrieval* shows a high ability to retrieve $\tau_{cw}$ with a correlation of $|r| = 0.98$, as well as to retrieve $CWP$ with $|r| = 0.95$ and $r_{total}$ with $|r| = 0.86$. Using the dataset from the campaign, a comparison between $CWP$ from *Total Cloud Water retrieval* and Cloudnet was performed and showed a correlation of $|r| = 0.81$. In conclusion, the comparison to artificial clouds and the validation using Cloudnet showed that *Total*

*Cloud Water retrieval* is able to retrieve the condensed water path from clouds for optically thin clouds and makes it a useful complementation for thin clouds to existing microwave-based measurements.

## 1 Introduction

Clouds play an important role in climate, due to their impact on the radiation budget. In the visible regime, clouds mainly
reflect solar radiation with respect to their albedo and prevent solar radiation from reaching the earth surface, whereas in the infrared regime clouds hinder longwave radiation from escaping to space and re-emit it back to earth, where it warms the





surface. In general, clouds and aerosols are assumed to give a negative contribution to the total radiative forcing, but the exact amount remains unclear (Myhre et al., 2013). A big challenge is the description of optically thin clouds with a liquid water path ($LWP$) of $LWP < 100\,\mathrm{g\cdot m^{-2}}$. In the Arctic, about $80\%$ of the liquid water containing clouds are below this threshold (Shupe and Intrieri, 2004). Up to around $LWP = 40\,\mathrm{g\cdot m^{-2}}$, a change in the water path of a cloud leads to a change in the longwave radiative flux. For $LWP > 40\,\mathrm{g\cdot m^{-2}}$, the longwave radiative flux becomes less sensitive to a change in the liquid water path (Turner et al., 2007). The blackbody-limit for downwelling flux is $LWP > 90\,\mathrm{g\cdot m^{-2}}$, whereas for upwelling flux there is still sensitivity for $LWP > 100\,\mathrm{g\cdot m^{-2}}$, due to multiple scattering (Rowe et al., 2013).

In the Arctic, a much faster warming than on the rest of the earth takes places, called *Arctic amplification* (Wendisch et al., 2019). A larger number of processes are known to influence these Arctic amplification, but the quantification of each process and its importance is difficult. The project Arctic Amplification: Climate Relevant Atmospheric and Surface Processes and Feedback Mechanisms $(\mathcal{AC})^3$ (Wendisch et al., 2019) aims to close this gap of knowledge by performung various campaigns, model studies and enduring measurements in the Arctic. The measurement campaign presented in this paper is part of $(\mathcal{AC})^3$. Usually microwave radiometer (MWR) are used for ground-based observations of liquid water clouds. Their advantage is a high range of detectable liquid water, also they have the ability to operatore continiously 24 hours a day, but MWR suffer a high uncertainty in the $LWP$ of at least $15\,\mathrm{g\cdot m^{-2}}$ (Turner et al., 2007). Also MWR usually are only able to measure liquid water. For more accurate observations of optically thin clouds, *Fourier Transform Infrared (FTIR) Spectrometer* can be used to supplement existing cloud observation techniques.

Absortion FTIR spectrometers using the sun as light source are widely used for the observations of trace gases in the near-infrared (*Total Carbon Column Observation Network, TCCON* and *COllaborative Carbon Column Observing Network, COC-CON*) and mid-infrared (*Network for the Detection of Atmospheric Composition Change - Infrared working group, NDACC-IRWG*) region. Emission measurements can be used for the observation of trace gases in absence of the sun or the moon as light source, done for example by Becker et al. (1999) and Becker and Notholt (2000), as well as for the observation of optically thin clouds, performed within the scope of the network of the Atmospheric Radiation Measurement (ARM) using Atmospheric Emitted Radiance Interferometer (AERI). These spectrometers are built in particular for emission measurements. An emission FTIR spectrometer with higher spectral resolution compared to AERI has been set up on the German research vessel Polarsten to perform measurements in summer 2017 in the Arctic around Svalbard.

To invert the spectral radiances to retrieve microphysical cloud parameters, a new retrieval algorithm, called *Total Cloud Water retrieval* (TCWret), has been developed explicitly for this measurement campaign. TCWret uses the spectral radiances from $770.9\,\mathrm{cm^{-1}}$ to $1163.4\,\mathrm{cm^{-1}}$, where low absorption of gases occur and therefore the atmosphere is transparent for emissions from clouds. The aim of TCWret is to retrieve the cloud water optical depth $\tau_{cw} = \tau_{liquid} + \tau_{ice}$, the total effective droplet radius $r_{total}$ and the condensed water path $CWP = LWP + IWP$ ($IWP$: Ice Water Path) of Arctic mixed-phase clouds using an optimal estimation algorithm (Rodgers, 2000). The principle of this retrieval technique has been proven already for mixed-phase clouds using the far infrared and thermal spectral range (mixed-phase cloud property retrieval algorithm *MIXCRA* by Turner (2005) and CLoud and Atmospheric Radiation Retrieval Algorithm *CLARRA* by Rowe et al. (2019)) and for single-phase liquid clouds using the thermal infrared spectral range (extended line-by-line atmospheric transmittance and radiance





algorithm *XTRA* by Rathke and Fischer (2000) and with a statistical retrieval by Marke et al. (2016)).

Section 2 describes the measurement setup and the measurement area. This is followed by an explanation of the working principle of the retrieval TCWret in section 3. Section 4 describes the datasets used for testing and validation. Section 5 shows the performance of TCWret on the retrieval of microphysical cloud parameters from synthetic testcases and the comparison of the results from TCWret with results from CLARRA and Cloudnet. Finally, a summary and conlusion is provided.

## 2 Measurements

### 2.1 Area of measurements

Measurements were performed around Svalbard in summer 2017 from the 24th May 2017 until the 19th August 2017. The measurements with the FTIR were part of the legs PS106.1 (PASCAL), PS106.2 (SiPCA) and PS107 (FRAM), performed by the research vessel Polarstern. For further description see Macke and Flores (2018) and Schewe (2018). Figure (1) shows the positions of the measurement sites and the positions of the ship.

### 2.2 Setup of FTIR measurements

Measurements were performed using a mobile **F**ourier **T**ransform **I**nfra**R**ed (FTIR) spectrometer, the Equinox 55 by Bruker. The interferometer inside the instrument has a movable mirror which gives a maximum optical path difference of $3\,\mathrm{cm}$, which results in a maximum spectral resolution of $\Delta\bar{\nu} = 0.3\,\mathrm{cm}^{-1}$. The diameter of the entrance aperature was chosen, that a maximum illumination of the detector with atmospheric radiation without any contribution from the entrance aperature was attained. In this setup, the detector acts as aperture. As detecter, a MCT-detector was used. The spectrometer was placed in an air conditioned container on the *RV Polarstern*, which was located on the top deck of the ship. The setup was in zenith geometry. To get the spectral radiances from the spectra, a total power calibration has to be performed (Revercomb et al., 1988). Every measurement of the emission of the atmosphere consisted of three consecutive measurements: two measurements of the blackbody, $I_{hot}$ and $I_{amb}$, respectively and one measurement pointing skywards, $I_{atm}$. From

$$L_{atm} = B_{\bar{\nu}}(T_{amb}) + \frac{B_{\bar{\nu}}(T_{hot}) - B_{\bar{\nu}}(T_{amb})}{\mathcal{F}(I_{hot} - I_{amb})} \cdot \mathcal{F}(I_{atm} - I_{amb}) \tag{1}$$

the spectral radiance of the atmosphere can be calculated. $B_{\bar{\nu}}(T_{hot,amb})$ are the Planck functions for the hot temperature and for the ambient temperature and $I_{atm,hot,amb}$ are the interferograms for the atmospheric radiation, the black body radiator at high temperature and the black body radiator at ambient temperature. $\mathcal{F}$ is the operator for the Fourier transform, so $\mathcal{F}(I)$ are the spectra with arbitrary unit. In constrast to Revercomb et al. (1988), first the difference between the interferograms has been calculated. The Fourier transform has been applied to the difference of the interferograms. The unit of the resulting spectrum $L_{atm}$ is $\mathrm{mW} \cdot (\mathrm{sr} \cdot \mathrm{m}^2 \cdot \mathrm{cm}^{-1})^{-1}$.

Figure (2) shows measured spectral radiances for an optically thin cloud and for an optically thicker cloud in the spectral region between $770\,\mathrm{cm}^{-1}$ and $1200\,\mathrm{cm}^{-1}$.





## 2.3 Cloudnet measurements

The OCEANET-Atmosphere observatory from the Leibniz Institute for Tropospheric Research (TROPOS) in Leipzig (Germany) performed continuous measurements during PS106.1 and PS106.2 (Griesche et al., 2019). Its container houses a multi-
wavelength Raman polarization lidar Polly-XT and a microwave radiometer HATPRO which was complemented during PS106 by a vertically-pointing motion-stabilized 35-GHz cloud radar Mira-35. The OCEANET measurements provide profiles of aerosol and cloud properties and column-integrated liquid water and water vapor content. To retrieve products like liquid and ice water content the instrument synergistic approach Cloudnet (Illingworth et al., 2007) was applied to these observations.

## 3 Total Cloud Water retrieval (TCWret)

**T**otal **C**loud **W**ater **ret**rieval (TCWret) is a retrieval for microphysical cloud parameters from FTIR spectra. It is inspired by MIXCRA (Turner, 2005), CLARRA (Rowe et al., 2019) and XTRA (Rathke and Fischer, 2000) and uses an optimal estimation approach (Rodgers, 2000) to invert the measured spectral radiances for retrieving microphysical cloud parameters. TCWret is a new software developed at the Institute of Environmental Physics (University of Bremen) and is originally designed to be applied to the dataset acquired during the PS106 and PS107.

### 3.1 Working principle of TCWret

Two radiative transfer models are used in TCWret: the Line-By-Line Radiative Transfer Model (LBLRTM) (Clough et al., 2005) and the DIScrete Ordinate Radiative Transfer model (DISORT) (Stamnes et al., 1988). LBLRTM calculates the optical depth for gaseous absorbers and the water vapour continuum. The profiles of $H_2O$, $CO_2$, $O_3$, $CO$, $CH_4$ and $N_2O$ either can be set by the user, or a predefined atmosphere of LBLRTM can be used.
DISORT calculates the monochromatic radiative transfer through an inhomogenious plane-parallel medium including scattering, absorption and emission. DISORT calculates the spectral radiances by utilizing the optical depths from LBLRTM and the optical depths and effective droplet radii of a cloud. The present setup uses 16 streams in the calculation, which means, 16 differential equations with different polar angles for the intensity are solved (Stamnes et al., 2000).
Both models are coupled using LBLDIS (Turner, 2005). In LBLDIS, Optical depths and effective droplet radii are separately
stated for water and ice clouds. In the present setup of TCWret, the droplet size distribution follows a gamma size distribution (Turner et al., 2003). Single scattering parameters of liquid and ice water droplets were created by (Turner, 2014) and taken from the LBLDIS model. Scattering properties for spherical droplets have been calculated using the Mie algorithm by Wiscombe (1980). Refractive indices for liquid water were taken from Downing and Williams (1975). Temperature dependend refractive indices for the temperatures $240\,K$, $253\,K$ and $263\,K$ were taken from Zasetsky et al. (2005). Because single
scattering parameters for LBLDIS need to be in the format provided by Turner (2014), single scattering parameters from the temperature-dependent complex refractive indices (CRI) of Zasetsky et al. (2005) were used. However, it is important to note that they have large uncertainties from $1000\,cm^{-1}$ to $1300\,cm^{-1}$ (Rowe et al., 2013). For ice water droplets, refractive indices





of Warren (1984) are used. Scattering properties for more complex ice particle shapes like aggregates, bullet rosettes, droxtals, hollow columns, solid columns, plates and spheroids were calculated by Yang et al. (2013).

Spectral radiances are divided into microwindows. Microwindows shown in table (1) are chosen according to Rowe et al. (2019) and modified. Cloud parameters are retrieved from the offset, slope and curvature of the spectral radiance in the spectrum, so the emision lines of the trace gases distort the retrieval, if the amount of the trace gases is unknown. The usage of microwindows allows to minimize the impact of the unknown amount of trace gases by their lines in the spectra.

TCWret retrieves the parameter vector $\boldsymbol{x} = (\tau_{liquid}, \tau_{ice}, r_{liquid}, r_{ice})$, containing the optical depths of liquid water and ice
water and the effective droplet radii of liquid water droplets and ice water droplets, from a measured spectrum. The retrieval of microphysical cloud parameters is a nonlinear problem, so an iterative algorithm is needed:

$$\boldsymbol{x}_{n+1} = \boldsymbol{x}_n + \boldsymbol{s}_n \tag{2}$$

Here $\boldsymbol{x_n}$ and $\boldsymbol{x_{n+1}}$ are the cloud parameters of the $n$-th and $(n+1)$-th step and $\boldsymbol{s_n}$ is the modification of the cloud parameters during the $n$-th iteration. To determine the adjustment vector $\boldsymbol{s_n}$, an *optimal estimation* approach has been chosen (Rodgers,
2000). The governing equation is

$$\left(\mathbf{K}^T \mathbf{S_y}^{-1} \mathbf{K}_n + \mathbf{S_a}^{-1} + \mu^2 \mathbf{S_a}^{-1}\right) \boldsymbol{s}_n = \mathbf{K}_n^T \mathbf{S_y}^{-1} \left[\boldsymbol{y} - F(\boldsymbol{x}_n)\right] + \mathbf{S_a}^{-1} \cdot (\boldsymbol{x_a} - \boldsymbol{x}_n) \tag{3}$$

The quantities in the equation are the jacobian matrix $\mathbf{K} = \left(\frac{\partial F(x_i)_j}{\partial x_i}\right)$, a weighting matrix $\mathbf{S_y}^{-1} = \mathrm{diag}(\sigma_i^{-1})$ containing the variances of the spectral radiance, the a priori $\boldsymbol{x_a}$ and the inverse error of the a priori $\mathbf{S_a}^{-1}$, the measured spectral radiances $\boldsymbol{y}$, the calculated spectral radiances $F(\boldsymbol{x_n})$ and the Levenberg-Marquardt term $\mu^2 \cdot \mathbf{S_a}^{-1}$. If the Levenberg-Marquardt term equals
0, then the retrieval uses a Gauss-Newton algorithm, otherwise a mixture of Gauss Newton and steepest descent ($\mu$ large) is used. If the cost function $\xi^2$ decreases, $\mu$ decreases as well. The aim of the iterations is to minimize the cost function $\xi^2(\boldsymbol{x})$.

$$\xi^2(\boldsymbol{x}_n) = \left[\boldsymbol{y} - F(\boldsymbol{x}_n)\right]^T \mathbf{S_y}^{-1} \left[\boldsymbol{y} - F(\boldsymbol{x}_n)\right] + \left[\boldsymbol{x_a} - \boldsymbol{x}_n\right]^T \mathbf{S_a}^{-1} \left[\boldsymbol{x_a} - \boldsymbol{x}_n\right] \tag{4}$$

Convergence is reached, if the change of the costfunction is below a given threshold, here set to $0.1\%$:

$$\frac{\xi^2(x_{n+1}^2) - \xi^2(x_n)}{\xi^2(x_{n+1})} < 0.001 \tag{5}$$

An important quantity to characterize the retrieval quality is the *Averaging Kernel Matrix* $\mathbf{A_r}$. The averaging kernel matrix contains the derivatives of the retrieved quantities with respect to the true state vector

$$\mathbf{A}_r = \frac{\partial \boldsymbol{x_r}}{\partial \boldsymbol{x_t}} \tag{6}$$

On the diagonal elements one finds the derivatives of each element in the retrieved state vector with respect to its corresponding element in the true state vector. Averaging kernels for the Levenberg-Marquardt algorithm are calculated via

$$\mathbf{A}_r = \mathbf{T}_r \mathbf{K}_r \tag{7}$$





with

$$\mathbf{T}_0 = \mathbf{0} \tag{8}$$

$$\mathbf{T}_{n+1} = \mathbf{G}_n + (\mathbf{I} - \mathbf{G}_n\mathbf{K}_n - \mathbf{M}_n\mathbf{S_a}^{-1})\mathbf{T}_n \tag{9}$$

$$\mathbf{G}_n = \mathbf{M}_n\mathbf{K}_n^T\mathbf{S_y}^{-1} \tag{10}$$

$$\mathbf{M}_n = \left(\mathbf{K}_n^T\mathbf{S_y}^{-1}\mathbf{K}_n + \mathbf{S_a}^{-1} + \mu^2\mathbf{D}_n\right)^{-1} \tag{11}$$

following Ceccherini and Ridolfi (2010). $\mathbf{T}_r$ is the final transfer matrix $\mathbf{T}$.

To get a better initial guess for the optical depth, the spectral radiances will be calculated for different $\tau_{cw} = \tau_{liquid} + \tau_{ice}$ with $\tau_{ice} = \tau_{liquid}$. The $\tau_{cw}$ which leads to the smallest difference between measured spectral radiances and calculated spectral radiances will be used as initial guess and a priori for the iteration. Two examples for fitted spectra of one thin cloud and one thick cloud is shown in figure (3).

## 3.2   Products of TCWret

Direct retrieval products are $\tau_{liquid}$, $\tau_{ice}$, $r_{liquid}$ and $r_{ice}$. From these parameters the water paths are calculated:

$$LWP = \frac{2}{3} \cdot r_{liquid} \cdot \tau_{liquid} \cdot \varrho_{liquid} \tag{12}$$

$$IWP = \frac{\tau_{ice} \cdot V_0(r_{ice})}{ext(r_{ice})} \cdot \varrho_{ice} \tag{13}$$

$$CWP = LWP + IWP \tag{14}$$

with the volumetric mass densities of liquid water $\varrho_{liquid} = 1000\,\text{kg}\cdot\text{m}^{-3}$ and ice water $\varrho_{ice} = 916.896\,\text{kg}\cdot\text{m}^{-3}$, the volume of an ice droplet $V_0(r_{ice})$ and the extinction cross section of an ice droplet $ext(r_{ice})$. The formula for $LWP$ works for spherical droplet only, while the formula for $IWP$ is valid for ice droplets of any shape (Turner, 2005).

Assuming $V_0(r_{ice}) = c_{V_0} \cdot r_{ice}^3$ and $ext(r_{ice}) = c_{ext} \cdot r_{ice}^2$, the formula of $CWP$ can be rearranged, so it can be calculated from $\tau_{cw}$ and a second parameter, which will be interpreted as the weighted total effective droplet radius $r_{total}$:

$$CWP = \frac{2}{3} \cdot \varrho_{liquid} \cdot \tau_{cw} \cdot r_{total} \tag{15}$$

with

$$r_{total} = (1 - f_{ice}) \cdot r_{liquid} + \frac{3}{2} \cdot \frac{\varrho_{ice}}{\varrho_{liquid}} \cdot \frac{c_{V_0}}{c_{ext}} \cdot f_{ice} \cdot r_{ice}. \tag{16}$$

For example, for spherical ice droplets is $\frac{c_{V_0}}{c_{ext}} = 0.627$.

Without the spectral windows below $600\,\text{cm}^{-1}$, a phase determination is less reliable (Rathke et al., 2002), so from the available dataset only $\tau_{cw}$, $r_{total}$ and $CWP$ are assumed to be viable products of TCWret. The remaining products $f_{ice} = \tau_{ice} \cdot \tau_{cw}^{-1}$, $r_{liquid}$ and $r_{ice}$ will be retrieved and saved and can be used for diagnostic applications.





### 3.3 Error estimation

Error estimation for the parameters is performed using the variance-covariance matrix of the retrieved state vector $x_r$ (Ceccherini and Ridolfi, 2010). Using the final transfer matrix $\mathbf{T}_r$ and the covariance matrix of the measurements $\mathbf{S}_y$, the variance-covariance matrix of $x_r$ is calculated from

$$\mathbf{S}_r = \mathbf{T}_r \mathbf{S}_y \mathbf{T}_r^T \tag{17}$$

The diagonal elements of $\mathbf{S}_r$ are the variances of $x_r$. The standard deviation of $CWP$ and $r_{total}$ will be calculated using error propagation. Alternatively, the errors can be calculated from the residuum. Here the maximum absolute value of the residuum is considered to be the standard deviation and the covariance-matrix of the measurement is set to $\mathbf{S}_y = \sigma_{residuum}^2 \cdot \mathbf{I}$.

## 4 Datasets

### 4.1 Synthetic testcases

A set of synthetic testcases containing spectral radiances of artifical clouds with known cloud parameters, created by Cox et al. (2016) will be used to test the ability of TCWret to retrieve $\tau_{cw}$, $r_{total}$ and $CWP$. Several different types of simulated clouds are available in this dataset:

- Singlelayer Clouds: The entire cloud is assumed to be in one altitude model layer.

- Multilayer Clouds: The artifical cloud extends over a larger number of model layers.

- Thin boundaries: Clouds with inhomogenious distribution of cloud water.

- Liquid topped: The cloud is topped by a liquid water layer.

- Different ice habit: Ice droplets are not spheres anymore, but plates, hollow columns or solid columns.

Ice droplets are set to spherical in TCWret for all cases. $\tau_{cw}$, $r_{total}$ and $CWP$ of these testcases are retrieved under different assumptions:

- Added noise: A random noise of $0.2\,\mathrm{mW} \cdot (\mathrm{sr} \cdot \mathrm{m}^2 \cdot \mathrm{cm}^{-1})^{-1}$ is added to the spectral radiances.

- Temperature offset: The temperature profile is disturbed with a constant offset of either $+1\,\mathrm{K}$ or $+5\,\mathrm{K}$.

- Radiance offset: A constant offset of $-2.0\,\mathrm{mW} \cdot (\mathrm{sr} \cdot \mathrm{m}^2 \cdot \mathrm{cm}^{-1})^{-1}$ is added to the spectral radiances.

These testcases cover common sources of errors: Interpolation of the atmospheric state can lead to errors in the atmospheric profiles. An error in the total power calibration like wrong temperature of the black body can lead to an offset in the spectral radiances. Noise of $0.2\,\mathrm{mW} \cdot (\mathrm{sr} \cdot \mathrm{m}^2 \cdot \mathrm{cm}^{-1})^{-1}$ is a typical value observed in the measurements from the campaign. Each of these disturbances is applied to the testcases solely.





## 4.2 IR-Retrievals

### 4.3 Data retrieved using TCWret

Profiles of temperature, humidity and pressure are taken from ECMWF analysis. Informations of the cloud height are taken from the Vaisala CL51, mounted on the *RV Polarstern* (Schmithüsen 2017a, b, c). Only the cloud base height is provided by the ceilometer, thus TCWret assumes clouds to be located at the cloud base layer. This choice has been made, because no information about the cloud top height is available for the PS107, so a consistent dataset of the entire campaign was created. Ice droplets were assumed to be spherical. 98.50% of the results have total cloud optical depths of $\tau_{cw} < 6.0$ with a maximum of 25.11% of the results in the interval $\tau_{cw} = [3.0, 4.0]$. 95.35% of the cases have $r_{total} < 20.0\,\mu m$, with a maximum of results in the interval from $10\,\mu m$ and $15\,\mu m$. 98.28% of the $CWP$ results are below $50\,g \cdot m^{-2}$. The highest number of results are in the interval $CWP = [20.0, 30.0]$ with 28.87% of the data. Statistics are shown in figure (4). Note that only optically thin clouds were measured, therefore these results do not give an average over all clouds present during the measurement campaign, but only about the thin clouds.

### 4.4 Data retrieved using CLARRA

For comparison, the spectral radiances measured during the campaign were analysed using the CLoud and Atmospheric Radiation Retrieval Algorithm (CLARRA) (Rowe et al., 2019). CLARRA uses optimal estmation to retrieve microphysical cloud parameters. Like in TCWret, CLARRA uses LBLRTM and DISORT as forward models, but the coupling between them is not done by LBLDIS, but with a newly developed coupling algorithm by Rowe et al. (2019). Results from CLARRA are restricted to cloud optical depths of $\tau_{cw} \in [0.25, 6.0]$. Complex refractive indices of Downing and Williams (1975) are used for liquid water. Ice particles were assumed to be spherical. Clouds are placed in the layer that is nearest to the height given by the ceilometer, which could result in slightly different cloud heights as used in TCWret. CLARRA uses the same microwindows as TCWret, including the windows from $558.0\,cm^{-1}$ to $562.0\,cm^{-1}$ and from $571.0\,cm^{-1}$ to $574.0\,cm^{-1}$.

### 4.5 Data from Cloudnet

The retrieved Cloudnet data set during PS106 has been made available via Pangaea (Griesche et al. 2020a, b). Liquid water path and ice water content are used to compute the $CWP$, which is compared to the retrieval of infrared spectra using TCWret.

#### 4.5.1 Ice Water Path

Cloudnet provides the ice water content, which needs to be converted into the ice water path by integration over the entire cloud.

$$IWP = \int_{z_{base}}^{z_{top}} IWC(z)\,dz. \tag{18}$$





$z_{base}$ and $z_{top}$ are the cloud boundaries given by Cloudnet and $IWC(z)$ is the ice water content. Errors of the $IWC$ are given in $\mathrm{dB}$ and converted into $\mathrm{g \cdot m^{-2}}$ by

$$\sigma_{IWC,g/m2} = \log_{10}(\sigma_{IWC,dB}). \tag{19}$$

Data flagged with *No ice*, *Reliable retrieval*, *Ice detected only by lidar* and *Would be identified as ice if below freezing* are incorporated.

### 4.5.2 Liquid Water Path

The liquid water path is taken directly from the Cloudnet retrievals. Data flagged with *No liquid water*, *Reliable retrieval*,
*Adiabatic retrieval: cloud top adjusted* and *Adiabatic retrieval: new cloud pixel* are incorporated. Data, where the integral of the liquid water contant does not equal the liquid water path are omitted from the comparison.

## 5 Results

First, the comparison of the retrieval results of the synthetic testcases to the known cloud parameters are shown. From these results abilities and limitation of TCWret are shown under incorporation of different errors. Then the retrievals of the PS106
and PS107 data performed by TCWret and CLARRA are shown. Finally the validation of the dataset from the PS106 retrieved using TCWret against the reference dataset retrieved using Cloudnet are presented and discussed.

### 5.1 TCWret vs. Testcases

Figure (5) shows the results for the undisturbed case. Figures (6), (7) and (8) show the results for the cases with added noise, radiance offset and temperature offset. The factors of over- and understimation given in this sections are the slopes of $x_{theory} =$
$a \cdot x_{retrieval}$ with $x$ as the corresponding cloud parameter. A full listing can be found in table (2).

#### 5.1.1 Undisturbed

Without any imposed error on the spectral radiances, TCWret shows a high correlation of $|r| = 0.98$ for $\tau_{cw} < 8$. However, for $\tau_{cw} > 6$ are less results available. Also from the spectra it can be seen, that for high optical depths of 6 and larger the response of the spectral radiance to a change in the cloud parameters strongly decreases, so results with $\tau_{cw} > 6$ are omitted from further
discussions. Correlation for the $CWP$ is $|r| = 0.91$ without any limitations of the maximum $CWP$. The condensed water path can retrieved without the exact knowledge of $f_{ice}$, $r_{liquid}$ or $r_{ice}$.
Retrievals for $r_{total}$ show a correlation of $|r| = 0.86$. As the $CWP$ is a product of $\tau_{cw}$ and $r_{total}$ (Equation 15), errors of $r_{total}$ propagate into $CWP$. A closer view at results where the retrieved $r_{liquid}$ is larger than $20\,\mu m$ shows, that a retrieved liquid effective droplet radius of $r_{liquid} > 20\,\mu m$ can serve as an indicator for inaccurate results of $r_{total}$ and $CWP$. Another
problem is the retrieval of ice droplets with shapes different from the assumed shape. Figure (9) shows results from cases with non-spherical ice parameters, without limitation of $r_{liquid}$. Even for low amounts of retrieved $CWP$ there is either an





underestimation or overestimation of up to one third. For higher amounts of $CWP$, a wrong ice shape leads to a wrong $CWP$ of up to a factor of 2. However, the retrieval of the ice shape is beyond the scope of this paper.

If the retrieved $r_{liquid}$ is limited to the empirically found value of $r_{liquid} < 20\,\mu m$, then the correlation for $CWP$ increases to

$|r| = 0.95$. The spread of the results decreases, indicated by the standard deviation in table (2).

### 5.1.2 Random noise

Retrievals of $\tau_{cw}$ from spectral radiances with added noise give a correlation of $|r| = 0.98$. Limiting $r_{liquid}$ to $20\,\mu m$, the correlation for $r_{total}$ is $|r| = 0.83$. For $CWP$, the correlation is $|r| = 0.95$. The retrieved $CWP$ overstimates the true $CWP$ by a factor of $0.97 \pm 0.01$.

### 5.1.3 Temperature offset

An offset in the temperature profile leads to a larger underestimation of $\tau_{cw}$. This understimation increases by increasing temperature offset. If the offset is $+1\,K$, then for $\tau_{cw} < 3$ the total cloud water optical depth is underestimated by a factor of $1.06 \pm 0.01$ with a correlation of $|r| = 0.98$. For $\tau_{cw} > 3$ the underestimation increases to $1.17 \pm 0.02$ with $|r| = 0.88$. If the temperature offset is $+5\,K$, then for $\tau_{cw} < 2$ the retrieved $\tau_{cw}$ underestimates the true $\tau_{cw}$ by a factor of $1.40 \pm 0.02$ ($|r| = 0.95$).

For $\tau_{cw} > 2$, a factor of $1.65 \pm 0.05$ for the underestimation of the retrieved $\tau_{cw}$ is found with a correlation of $|r| = 0.79$.

In the $+1\,K$-case, the correlation is $|r| = 0.80$ for $r_{total}$, whereas in the $+5\,K$-case it is $|r| = 0.75$. An offset of $+5\,K$ leads to a 5-fold overestimation at most, but this case benefits from flagging of the cases with $r_{liquid} > 20\,\mu m$.

There is no clear distinction for the $CWP$ as seen for $\tau_{cw}$. An offset of $+1\,K$ has little influence on the $CWP$ compared to the undisturbed case, as the overestimation retrieved $CWP$ is $0.93 \pm 0.01$ and the correlation remains high with $|r| = 0.95$. In the

$+5\,K$-case, the retrieved $CWP$ is underestimated by a factor of $1.16 \pm 0.02$ with a correlation of $|r| = 0.94$.

### 5.1.4 Radiance offset

A negative radiance offset is equivalent to a positive temperature offset. A lower spectral radiance is linked to a lower temperature of the cloud. But the temperature of the cloud remains the same, so temperature of the cloud is higher than it would be if the lowered spectral radiance are the true spectral radiances. This can be seen in the results of $\tau_{cw}$, which shows a similar

pattern as in the case of a positive temperature offset. $\tau_{cw}$ is underestimated by a factor of $1.19 \pm 0.01$ with a correlation of $|r| = 0.97$.

$r_{total}$ shows a correlation of $|r| = 0.62$, which is the lowest value of all the results in the error cases. In contrast to the $+5\,K$-case, the deviation of the results around the $1:1$-line increases more uniform, which also makes the $r_{liquid} > 20.0\,\mu m$-criterion less helpful.

The correlation for $CWP$ is $|r| = 0.75$. The outliers in figure (8.b) are cases with high ice effective droplet radii $r_{ice} = 61\,\mu m$. In the other cases, these results could be removed from the evaluation because of the criterion $r_{liquid} > 20\,\mu m$ for the $CWP$.





An offset in the spectral radiance imposes the largest errors. For the $CWP$, the standard deviation increases from $5.04\,\mathrm{g \cdot m^{-2}}$ in the undisturbed case up to $14.46\,\mathrm{g \cdot m^{-2}}$.

## 5.2 TCWret vs. CLARRA

Figure (10) shows the comparison of the results from TCWret and CLARRA for the cloud water optical depth $\tau_{cw}$, the total effective droplet radius $r_{total}$ and the condensed water path $CWP$. Although both retrievals use different coupling algorithms between LBLRTM and DISORT with different uncertainties in the spectral radiances (Rowe et al., 2019) and TCWret does not use the microwindows below $770.9\,\mathrm{cm^{-1}}$, all retrieved quantities show a high agreement between both retrievals. Thus, retrieval results of TCWret are consistent with those from CLARRA.

## 5.3 TCWret vs. Cloudnet

For the comparison between TCWret and Cloudnet, results from both datasets were averaged over a time period of three minutes. This has been done because the underlying measurement systems have different temporal resolutions, also both measurement systems were at different locations on the ship. A time interval of three minutes has been chosen, because it makes gaps in the timeseries less dominant, but still a variability due to moving clouds can be caught by the measurements.

Figure (11) shows the correlation between the retrieved $CWP$ of TCWret and Cloudnet. For TCWret retrievals with $\tau_{cw} < 6$ and $r_{liquid} < 20\,\mathrm{\mu m}$, the correlation between TCWret and Cloudnet is $|r| = 0.81$. Retrievals of Cloudnet show lower results than those of TCWret with a factor of $0.78$. Few datapoints are outside the standard deviations of the results. One reason for this is the usage of only the cloud base height from the ceilometer in cases were the cloud extends over a larger altitude range. In those cases, the true temperature of the cloud is different from the assumed temperature in the retrieval, which affects the

retrieved $\tau_{cw}$. For example, the result marked by the arrow in figure (11) shows one data point, where TCWret shows a much larger result outside the standard deviations of Cloudnet. If the entire cloud height information provided by Cloudnet is used, then the retrieved $CWP$ decreases by about $15\%$. Another source of error is the assumption of only spherical ice droplets. However, due to the lack of knowledge about the true ice shape, this source of error can hardly be reduced.

## 6 Summary

During summer 2017, measurements of infrared spectral radiance from clouds and atmosphere were performed using a FTIR spectrometer on the *RV Polarstern*. The campaigns PS106 and PS107 took place in the Arctic around Svalbard.

Total cloud optical depth $\tau_{cw}$, the total effective droplet radius $r_{total}$ and the condensed water path $CWP$ are retrieved from infrared spectra, measured using a FTIR spectrometer. The retrieval is performed using TCWret, which has been developed with focus on the present measurement campaign. TCWret uses an optimal estimation approach (Rodgers, 2000) with non-

linear Levenberg-Marquardt algorithm to fit simulated spectral radiance, calculated by LBLRTM and LBLDIS/DISORT.

Synthetic testcases with spectral radiances of artificial clouds are used to examine the ability of TCWret to retrieve $\tau_{cw}$ and $CWP$ with different errors like biases in the temperature profile or an offset in the measured spectral radiance. Retrievals





using these testcases show a general ability of TCWret to retrieve $\tau_{cw}$, $r_{total}$ and $CWP$. The ccorrelation of retrieved $\tau_{cw}$ and true $\tau_{cw}$ is between $|r| = 0.95$ and $|r| = 0.98$ in all cases. But imposing a larger temperature error of $+5\,\mathrm{K}$ or an offset

in the spectral radiance lead to an underestimation of the true $\tau_{cw}$. Retrievals of $CWP$ showed larger underestimations of the true $CWP$ if the retrieved $r_{liquid}$ is larger than $20\,\mu\mathrm{m}$. This finding is supported by the evaluation of $r_{total}$. In cases with $r_{liquid} > 20\,\mu\mathrm{m}$, the retrieved $r_{total}$ differs from the true $r_{total}$. Especially in the case with a temperature error of $+5\,\mathrm{K}$ the correlation increases from $0.49$ to $0.75$. Under assumption of random noise, the correlation is $|r| = 0.83$.

Without $r_{liquid} > 20\,\mu\mathrm{m}$, $CWP$ retrievals show a correlation between $|r| = 0.75$ and $|r| = 0.95$. The lowest correlation of

$|r| = 0.75$ occurs in the case of an added offset to the spectral radiances. In those cases the restriction of the retreived radius to $r_{liquid} < 20\,\mu\mathrm{m}$ is not sufficient anymore. An offset in the spectral radiance introduces the largest errors in $CWP$ retrievals. The standard deviation of the retrieved $CWP$ minus the true $CWP$ increases to $14.46\,\mathrm{g \cdot m^{-2}}$. Offsets in the spectral radiance can result from errors in the calibration like wrong temperatures of the black body radiator. Therefore, a careful calibration of the spectrometer is crucial for the retrieval of microphysical cloud parameters. Additionally, the wrong representation of ice

shapes in the retrieval leads to errors in $r_{total}$ and $CWP$, which can lead to a error of the $CWP$ from $33\%$ of the retrieved $CWP$ if $CWP < 40\,\mathrm{g \cdot m^{-2}}$ up to $200\%$, if $CWP > 60\,\mathrm{g \cdot m^{-2}}$. However, the retrieval of the ice particle shape is beyond the scope of this paper.

From the measurement campaign, 1808 sets of $\tau_{cw}$, $r_{total}$ and $CWP$ were retrieved using TCWret. $98.50\%$ of the results have $\tau_{cw} < 6.0$ and $95.35\%$ have $r_{total} < 20.0\,\mu\mathrm{m}$. $98.28\%$ of the results have $CWP < 50\,\mathrm{g \cdot m^{-2}}$.

A comparison to CLARRA from (Rowe et al., 2019) shows high agreement between retrieval results of both results, thus TCWret results are consistent with results from CLARRA.

Finally, TCWret has been validated against Cloudnet. Comparison of the $CWP$ retrievals shows a correlation of $|r| = 0.81$. Results from Cloudnet are lower than those of TCWret by a factor of $0.78$. Larger mismatches can either be explained by cloud height mismatches. If the cloud expands over larger altitudes, this induces errors in the temperature of the cloud and therefore

it distorts the retrieval. Another source of errors is the shape of the ice particles. However, retrievals of microphysical cloud parameters and cloud water paths by TCWret for thin clouds with low water amounts work well and results are in agreement with Cloudnet results.

## 7   Conclusions and Outlook

Measurements of infrared spectral radiance from clouds are a powerful instrument to acquire informations about the optical

depth and the water path of thin clouds with low amounts of water, which is an important complementation of microwave observations, that are more sensitive to thick clouds. Also the relatively simple retrieval with only the cloud base height known from ceilometer measurements gives consistent results for the cloud water path, as shown by TCWret in comparison to Cloudnet. Although retrievals using infrared spectral radiances can not be applied to optically thick clouds with larger optical depths than 6, TCWret can provide informations on the water path of optically thin clouds, where Cloudnet or pure microwave

radiometer measurements are less reliable due to the larger standard deviation. Best results are expected by a combination of



infrared and microwave retrievals (Marke et al., 2016).

The developed measurement setup with a mobile FTIR spectrometer and a black body radiator will be used to measure spectral radiance from clouds in Ny-Ålesund from winter 2017 onwards. The source code of TCWret is publicly available.

*Code and data availability.* Results for PS106 and PS107 retrieved by TCWret are available at https://doi.pangaea.de/10.1594/PANGAEA.
900377. Cloudnet retrieval products are available https://doi.pangaea.de/10.1594/PANGAEA.919452 and https://doi.pangaea.de/10.1594/
PANGAEA.899898. The latest version of TCWret can be downloaded from GitHub (https://github.com/RichterIUP/Total-Cloud-Water-retrieval).
The radiative transfer models LBLRTM (http://rtweb.aer.com/lblrtm.html) and DISORT (http://www.rtatmocn.com/disort/) and the coupling
algorithm LBLDIS (https://www.nssl.noaa.gov/users/dturner/public_html/lbldis/index.html) need to be downloaded separately.

*Author contributions.* Philipp Richter performed measurements during PS106 and PS107, implemented TCWret and retrieved microphysi-
cal cloud parameters using TCWret. Mathias Palm designed and built the measurement setup, performed measurements during the PS106.1
and gave advice in the development of TCWret. Christine Weinzierl performed measurements during the PS106.2 and built the measurement
setup. Hannes Griesche performed Cloudnet retrievals and gave advice in using the Cloudnet data. Penny Rowe performed CLARRA re-
trievals and gave advice in the application of the testcases. Justus Notholt gave advice in the setup of the measurement and the development
of TCWret. All authors contributed to manuscript revisions.

*Competing interests.* The authors declare no competing interests.

*Acknowledgements.* We gratefully acknowledge funding from the Deutsche Forschungsgemeinschaft (DFG, German Research Foundation,
TRR 172) - Projektnummer 268020496 - within the Transregional Collaborative Research Center - ArctiC Amplification: Climate Relevant
Atmospheric and SurfaCe Processes, and Feedback Mechanisms (AC)3 - in subproject B06 and E02. We thank the Alfred-Wegener-Institute
and RV Polarstern crew and captain for their support (AWI_PS106_00 and AWI_PS107_00). The computations were performed on the HPC
cluster Aether at the University of Bremen, financed by DFG within the scope of the Excellence Initiative.



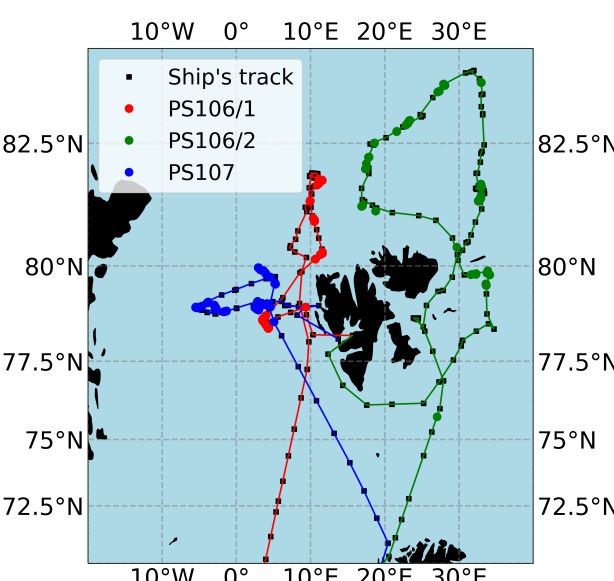

**Figure 1.** Map of the measurement area. Red dots indicate measurements during the PS106.1 (24th May 2017 until 21st June 2017), greens dots indicate measurements during the PS106.2 (23rd June 2017 until 19th July 2017) and blue dots indicate measurements during the PS107 (22nd July 2017 until 19th August 2017). Black markers show the position of the ship with a time gap of 6 hours between each marker.



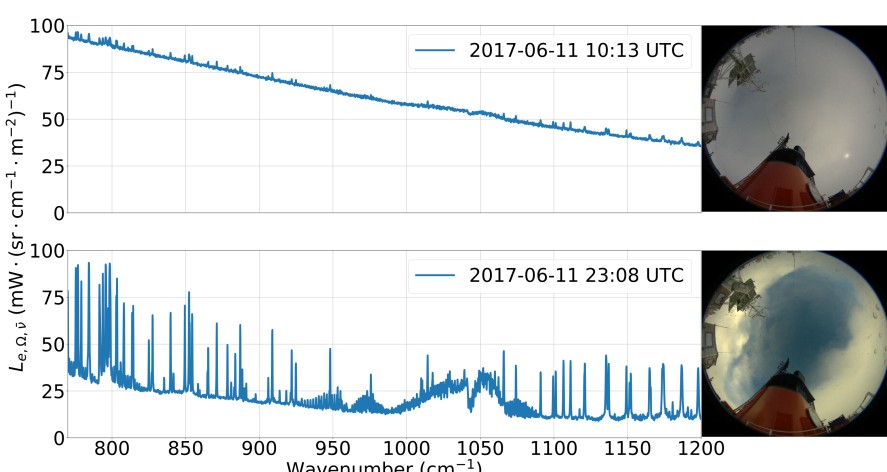

**Figure 2.** Spectral radiances measured by a FTIR spectrometer. Spectral radiances of the upper correspond to an optically thicker cloud, the ones of the lower plot are those from an optically thinner cloud.



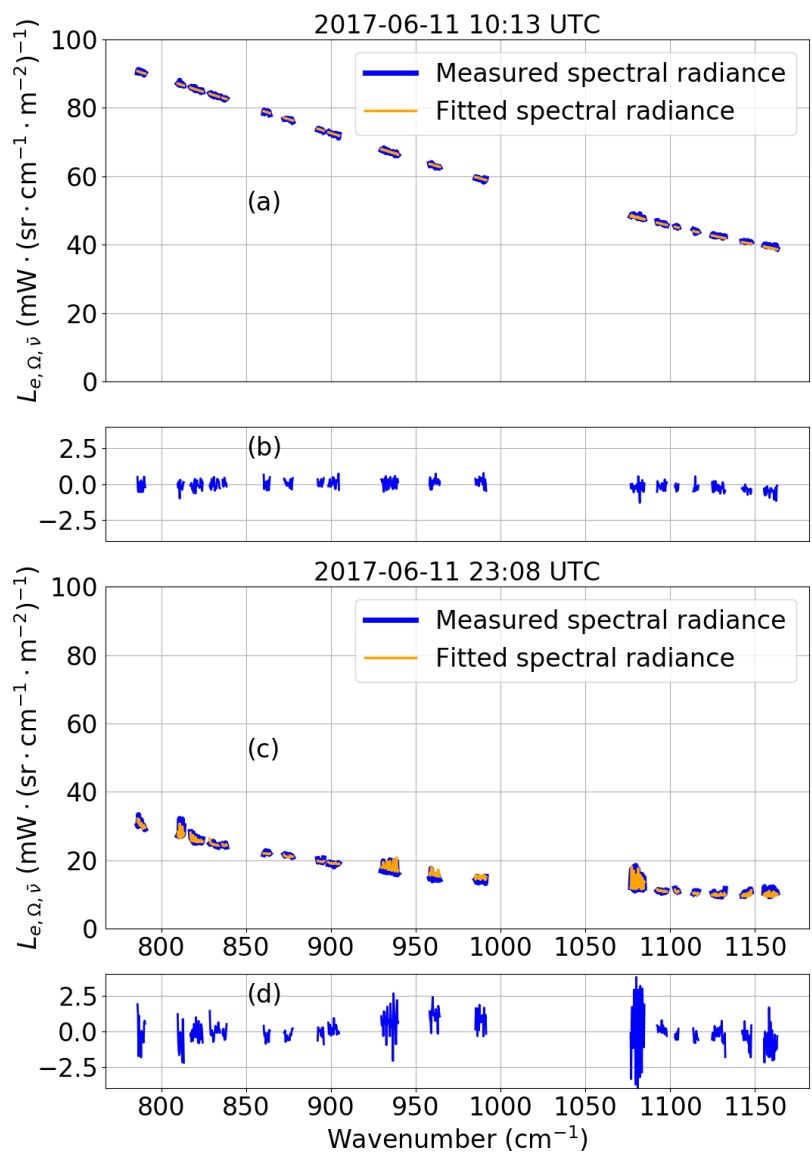

**Figure 3.** (a) and (b) are the spectral radiance and the residuum of an optically thick cloud, observed on the 11th June 2017, 10:13 UTC. (c) and (d) are the spectral radiance and the residuum of an optically thin cloud, observed on the same day, 23:08 UTC.

**Figure 4.** Plot (a), (c) and (e) show the relative amount of $\tau_{cw}$, $CWP$ and $r_{total}$, respectively. Plot (b), (d) and (f) show the corresponding cummulative sums.

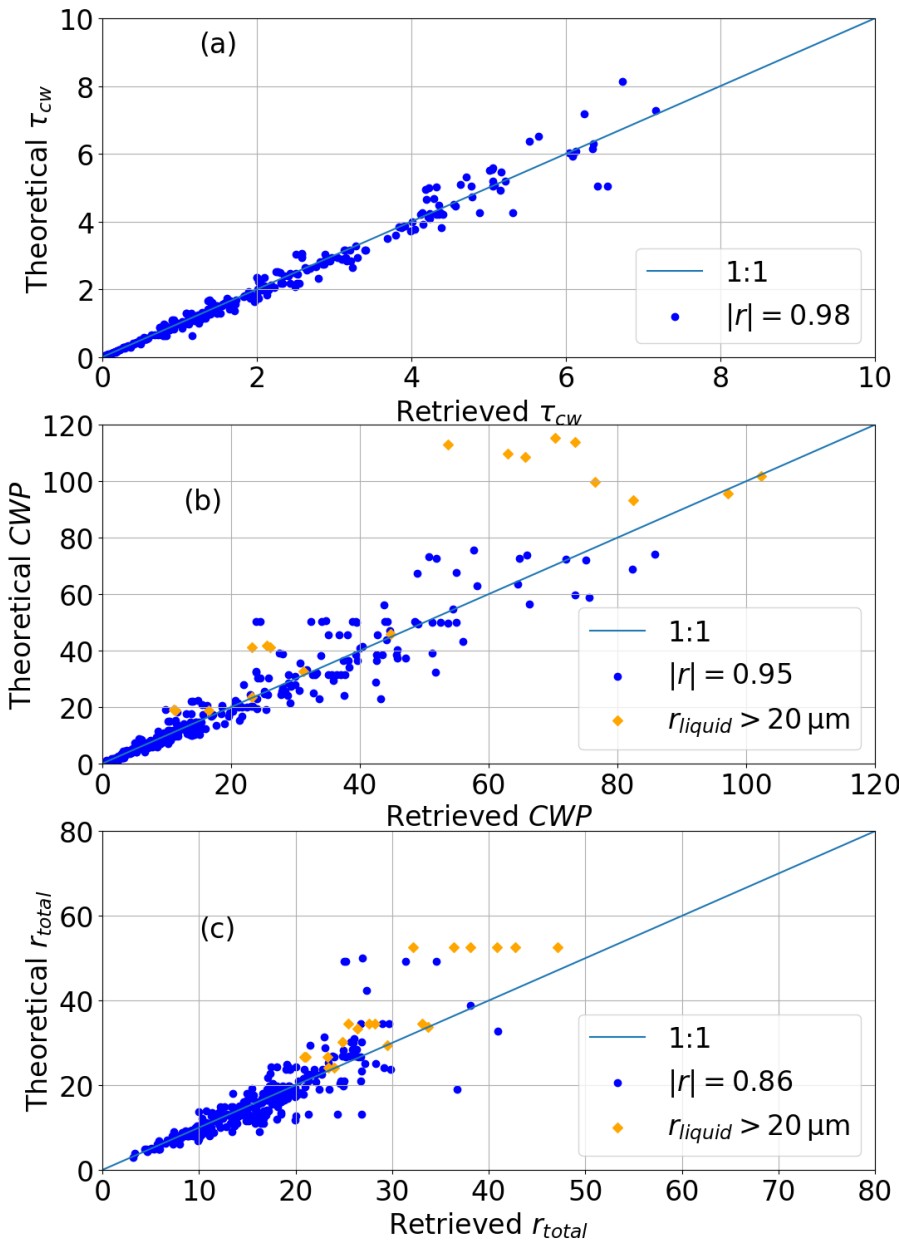

**Figure 5.** Comparison of retrieved results and true results for the undisturbed cases. Plot (a) shows the correlation for $\tau_{cw}$. Plot (b) shows the correlation for $CWP$. The orange markers show cases with retrieved $r_{liquid} > 20\,\mu\text{m}$. Plot (c) shows the retrieved $r_{total}$ versus true $r_{total}$. The orange markers show cases with retrieved $r_{liquid} > 20\,\mu\text{m}$.



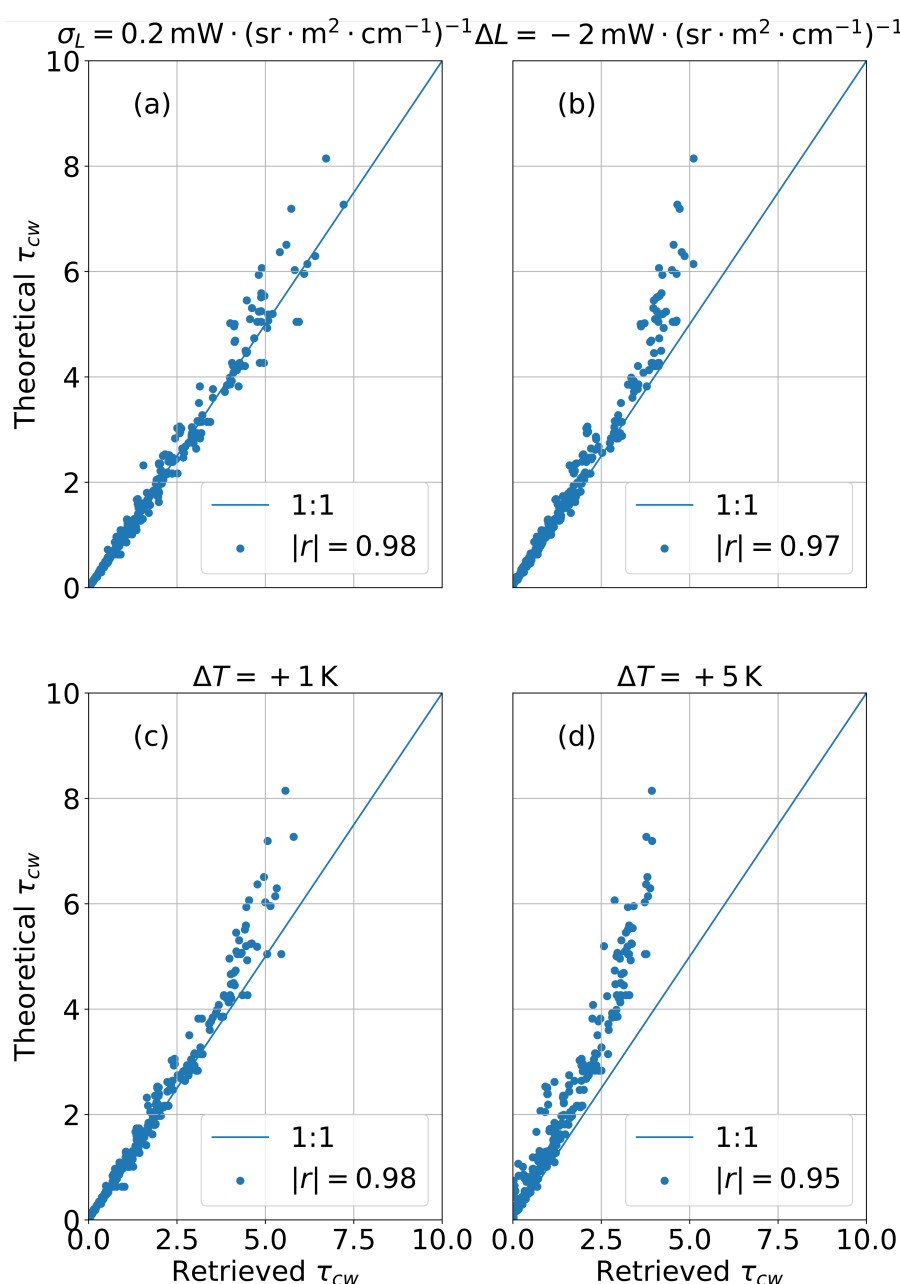

**Figure 6.** Comparison of retrieved $\tau_{cw}$ and theoretical $\tau_{cw}$ of testcases with different imposed errors. Plot (a) shows the correlation for the spectra with added random error of $0.2\,\mathrm{mW} \cdot (\mathrm{sr} \cdot \mathrm{m}^2 \cdot \mathrm{cm}^{-1})$ to the spectral radiances. Plot (b) shows the results for the cases with added offset of $-2.0\,\mathrm{mW} \cdot (\mathrm{sr} \cdot \mathrm{m}^2 \cdot \mathrm{cm}^{-1})$. In (c) and (d) an offset of $+1\,\mathrm{K}$ and $+5\,\mathrm{K}$ is added to the temperature profile.



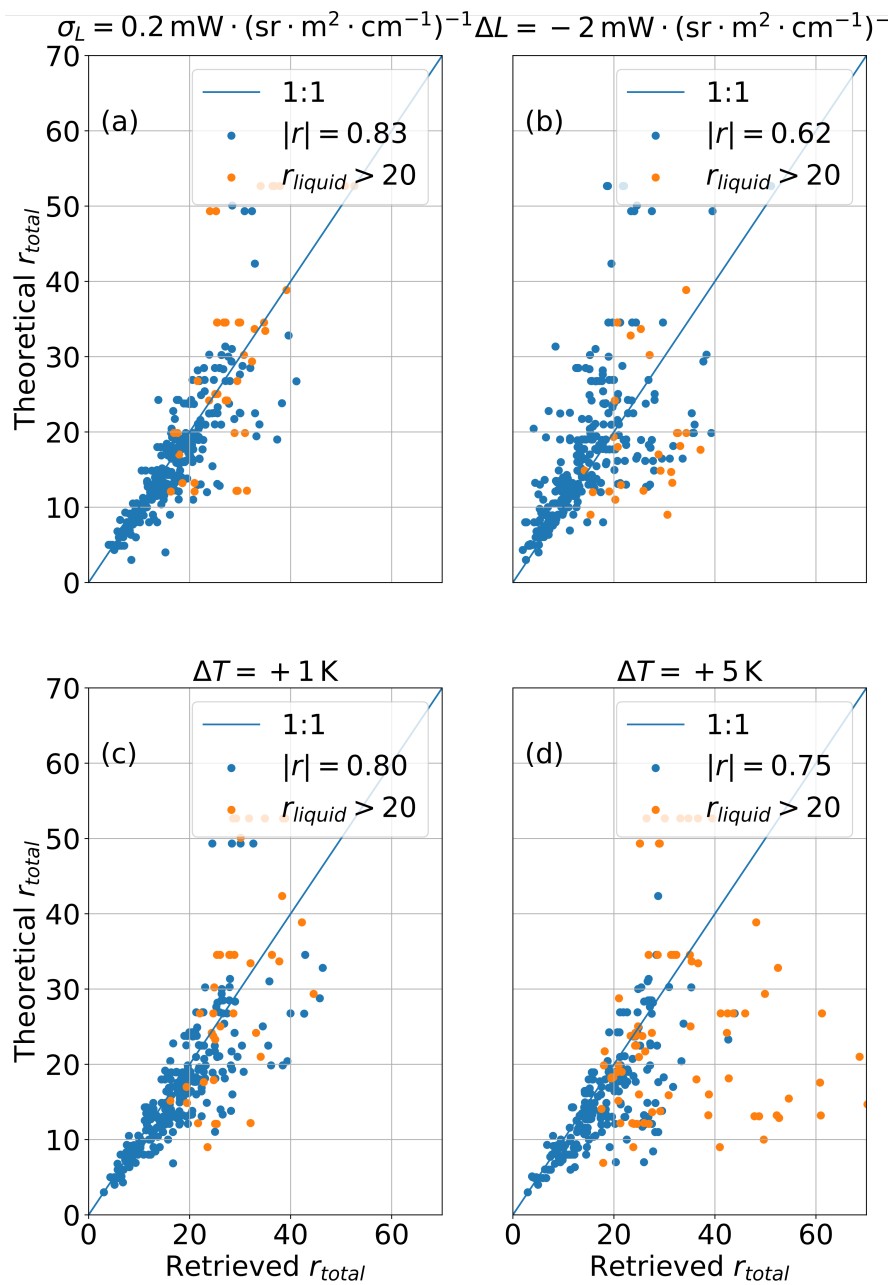

**Figure 7.** Comparison of retrieved $r_{total}$ and theoretical $r_{total}$ of testcases with different imposed errors. Plot (a) shows the correlation for the spectra with added random error of $0.2\,\mathrm{mW} \cdot (\mathrm{sr} \cdot \mathrm{m}^2 \cdot \mathrm{cm}^{-1})$ to the spectral radiances. Plot (b) shows the results for the cases with added offset of $-2.0\,\mathrm{mW} \cdot (\mathrm{sr} \cdot \mathrm{m}^2 \cdot \mathrm{cm}^{-1})$. In (c) and (d) an offset of $+1\,\mathrm{K}$ and $+5\,\mathrm{K}$ is added to the temperature profile. Orange markers in each plot show removed cases, due to $r_{total} > 20\,\mu\mathrm{m}$.



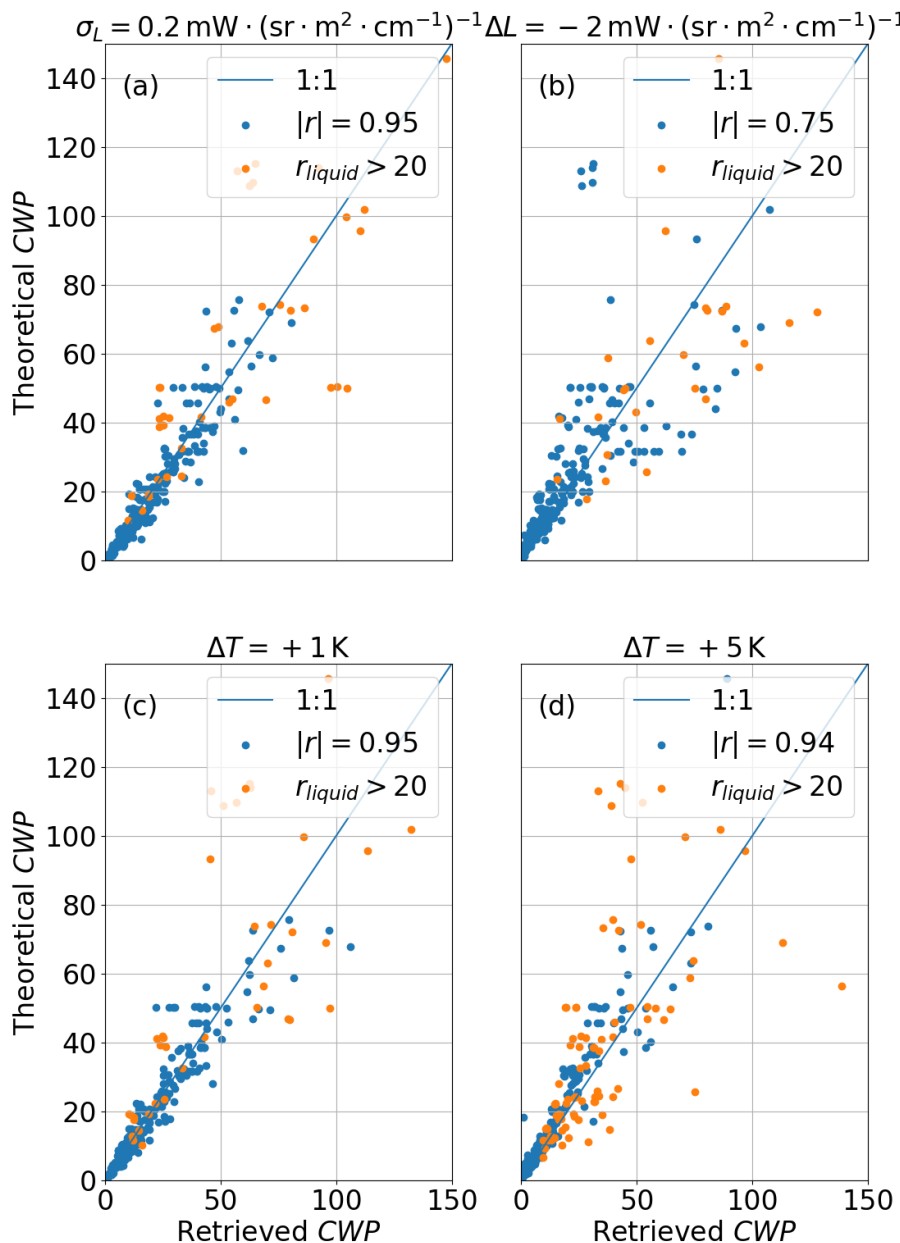

**Figure 8.** Comparison of retrieved $CWP$ and theoretical $CWP$ of testcases with different imposed errors. Results are retricted to values with $r_{liquid} < 20\,\mu$m. Plot (a) shows the correlation for the spectra with added random error of $0.2\,\text{mW} \cdot (\text{sr} \cdot \text{m}^2 \cdot \text{cm}^{-1})$ to the spectral radiances. Plot (b) shows the results for the cases with added offset of $-2.0\,\text{mW} \cdot (\text{sr} \cdot \text{m}^2 \cdot \text{cm}^{-1})$. In (c) and (d) an offset of $+1\,$K and $+5\,$K is added to the temperature profile. The outliers in plot (b) do not appear in the remaining plot, because they are removed due to the retricting of $r_{liquid} < 20\,\mu$m. Orange markers in each plot show removed cases, due to $r_{total} > 20\,\mu$m.



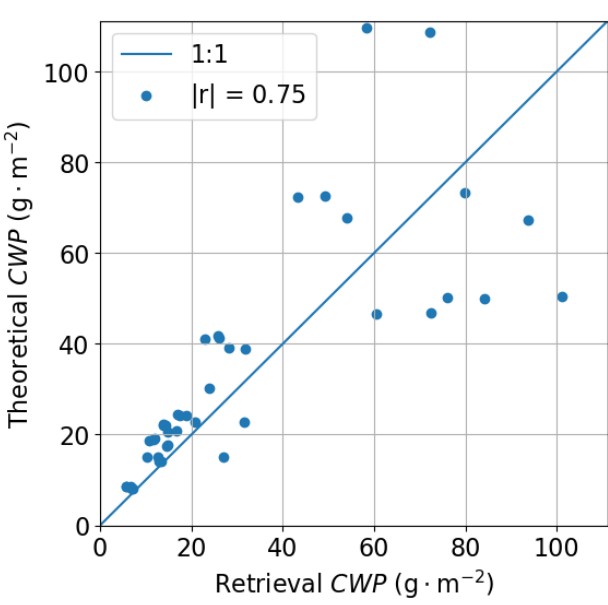

**Figure 9.** Retrieved $CWP$ and theoretical $CWP$, only for cases with non-spherical parameters, without limitation of $r_{liquid}$.



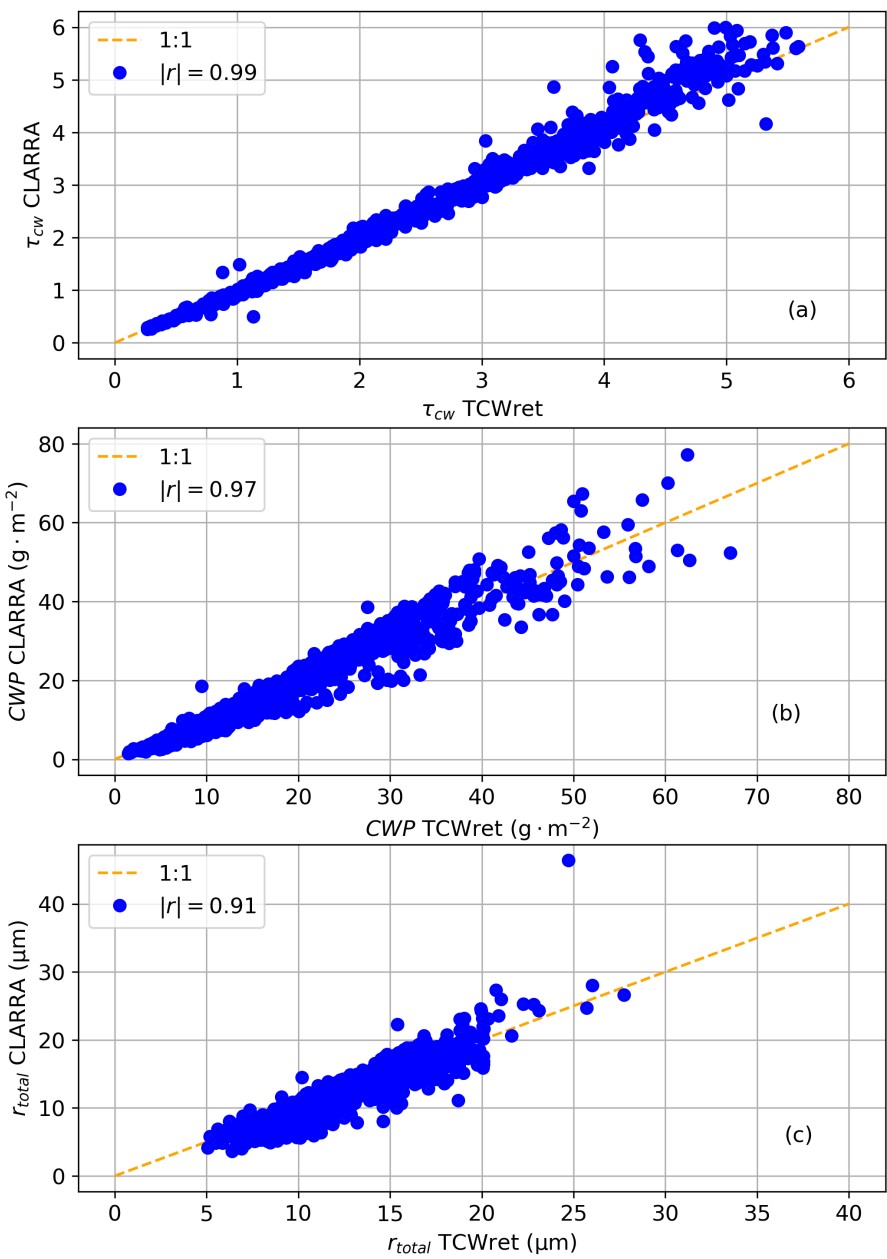

**Figure 10.** Comparison between TCWret and CLARRA for data from the PS106 and PS107. Plot (a) shows the correlation for $\tau_{cw}$. Plot (b) shows the correlation for $CWP$. Plot (c) shows the correlation for $r_{total}$.

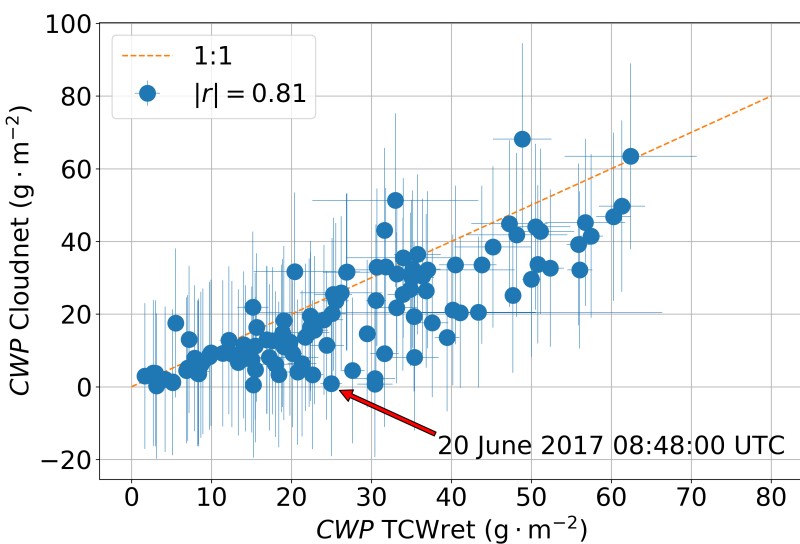

**Figure 11.** Comparison of $CWP$ between TCWret (x-axis) and Cloudnet (y-axis) for data from the PS106. Red arrow denotes the result from 2017-06-20 08:48 UTC. Errorbars show the standard deviations from TCWret and Cloudnet.





**Table 1.** Microwindows used in TCWret. Taken from Rowe et al. (2019)

| Lower wavenumber (cm$^{-1}$) | 785.9 | 809.5 | 817.0 | 828.3 | 835.8 | 860.1 | 872.2 | 891.9 | 898.2 |
| Upper wavenumber (cm$^{-1}$) | 790.7 | 813.5 | 824.4 | 834.6 | 838.7 | 864.0 | 877.5 | 895.8 | 904.8 |
| Lower wavenumber (cm$^{-1}$) | 929.6 | 958.0 | 985.0 | 1076.6 | 1092.2 | 1102.5 | 1113.3 | 1124.4 | 1142.2 | 1155.2 |
| Upper wavenumber (cm$^{-1}$) | 939.7 | 964.3 | 991.5 | 1084.8 | 1098.1 | 1105.0 | 1116.6 | 1132.6 | 1148.0 | 1163.4 |



**Table 2.** Coefficients for different retrieval cases.

| Case | Restrictions | $\|r\|$ | Mean | Standard deviation | Slope $a$: $x_{theory} = a \cdot x_{retrieval}$ |
|---|---|---|---|---|---|
| Undisturbed ($\tau_{cw}$) | | 0.98 | $-0.05$ | 0.37 | $1.05 \pm 0.01$ |
| Undisturbed ($r_{total}$) | $r_{total} < 20\,\mu m$ | 0.86 | 0.86 | 4.08 | $0.93 \pm 0.01$ |
| Undisturbed ($CWP$) | $r_{liquid} < 20\,\mu m$ | 0.95 | 0.24 | 5.04 | $0.99 \pm 0.01$ |
| Random noise ($\tau_{cw}$) | | 0.98 | $-0.05$ | 0.35 | $1.04 \pm 0.01$ |
| Random noise ($r_{total}$) | $r_{total} < 20\,\mu m$ | 0.83 | 0.97 | 4.28 | $0.93 \pm 0.01$ |
| Random noise ($CWP$) | $r_{liquid} < 20\,\mu m$ | 0.95 | 0.53 | 5.37 | $0.97 \pm 0.01$ |
| Radiance offset ($\tau_{cw}$) | | 0.97 | $-0.28$ | 0.54 | $1.19 \pm 0.01$ |
| Radiance offset ($r_{total}$) | $r_{total} < 20\,\mu m$ | 0.62 | $-2.50$ | 7.81 | $1.08 \pm 0.03$ |
| Radiance offset ($CWP$) | $r_{liquid} < 20\,\mu m$ | 0.75 | $-2.73$ | 14.46 | $0.95 \pm 0.03$ |
| Temperature $+1\,\mathrm{K}$ ($\tau_{cw}$) | | 0.98 | $-0.19$ | 0.46 | $1.14 \pm 0.01$ |
| Temperature $+1\,\mathrm{K}$ ($\tau_{cw}$) | $\tau_{cw} < 3$ | 0.98 | $-0.08$ | 0.16 | $1.06 \pm 0.01$ |
| Temperature $+1\,\mathrm{K}$ ($\tau_{cw}$) | $\tau_{cw} > 3$ | 0.88 | $-0.67$ | 0.84 | $1.17 \pm 0.02$ |
| Temperature $+1\,\mathrm{K}$ ($r_{total}$) | $r_{total} < 20\,\mu m$ | 0.80 | 1.63 | 4.97 | $0.88 \pm 0.01$ |
| Temperature $+1\,\mathrm{K}$ ($CWP$) | $r_{liquid} < 20\,\mu m$ | 0.95 | 0.44 | 5.71 | $0.93 \pm 0.01$ |
| Temperature $+5\,\mathrm{K}$ ($\tau_{cw}$) | | 0.95 | $-0.66$ | 0.80 | $1.51 \pm 0.02$ |
| Temperature $+5\,\mathrm{K}$ ($\tau_{cw}$) | $\tau_{cw} < 2$ | 0.95 | $-0.46$ | 0.47 | $1.40 \pm 0.02$ |
| Temperature $+5\,\mathrm{K}$ ($\tau_{cw}$) | $\tau_{cw} > 2$ | 0.79 | $-2.14$ | 1.13 | $1.65 \pm 0.05$ |
| Temperature $+5\,\mathrm{K}$ ($r_{total}$) | $r_{total} < 20\,\mu m$ | 0.75 | 2.72 | 5.01 | $0.82 \pm 0.02$ |
| Temperature $+5\,\mathrm{K}$ ($CWP$) | $r_{liquid} < 20\,\mu m$ | 0.94 | $-2.96$ | 6.60 | $1.16 \pm 0.02$ |



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
