# Peer review of "Retrieval of microphysical cloud parameters from EM-FTIR spectra measured in Arctic summer 2017"

_Atmospheric Measurement Techniques, 2020_

## Referee Comment (RC1) · Anonymous Referee #1 · 18 Sep 2020

General comments:

Using information derived from infrared spectral radiances, Richter et al. developed a new method for retrieving microphysical cloud parameters that characterize optically thin clouds. The method is designed to retrieve cloud water optical depth, total effective droplet radius and condensed water path. The work to compare and validate the retrieval results against different measurements has been carefully and systematically done. The exact amount of total radiative forcing due to clouds remains unclear, therefore additional measurement techniques providing complementary information such as the one presented by Richter et al. are important to shed light on this mechanism. I

recommend its publication in AMT after the questions, issues and comments outlined below have been addressed.

Specific comments:

Spectrometer: There is not much known to the reader about the Bruker Equinox 55. At least a chapter dedicated to the characterization of this instrument (instrument line shape, modulation efficiency, etc.), is necessary in my view.

Averaging Kernels: There is a discussion on the averaging kernels (AVK), however a presentation of the AVKs of the retrieval itself is lacking. It is not immediately clear to me how much information is coming from the Equinox 55 measurements and how much from the prior information. Therefore, I think it would benefit the manuscript to expand on this topic, show plots of AVKs, etc.

Error Analysis: The correlation plots are clear and helpful, but my impression is that the analysis on the error budget could still be improved for the reader. Specifically, I would have liked to see a table outlining the contributions of several variables to the total error budget.

Technical corrections:

P1, Line 15: "Cloud Water retrieval is able to retrieve the condensed water path from clouds for optically thin clouds" change to: "Cloud Water retrieval is able to retrieve the condensed water path from optically thin clouds" Or to: "Cloud Water retrieval is able to retrieve the condensed water path from clouds that are optically thin"

P2, Line 29: "takes places" –> takes place

P2, Line 32: "performung various campaigns" –> performing various campaigns.

P2, Line 34: "Usually microwave radiometer" –> Usually, microwave radiometers

P2, Line 35: "also they have the ability to operatore continiously 24 hours a day" –> also, they can continuously operate 24 hours a day

P2, Line 43: Notholt et al, 2000 missing in references

P3, Line 64: Please indicate coordinates of Svalbard

P4, Line 109: LBLDIS has not yet been defined

P4, Line 103: "either can be set" –> can either be set

P4, Line 113: "Temperature dependend" –> Temperature dependent

P6, Line 169: Please remove "is" in "For example, for spherical ice droplets is"

P8, Line 202: "Informations" –> Information

P8, Line 208: "interval from" –> interval between

P8, Line 211: I suggest changing "but only about the thin clouds" –> but only that of thin clouds

P9, Line 238-239: Please add comma and change "abilities and limitation" to "the capabilities and limitations", i.e. "From these results abilities and limitation . . ." –> "From these results, the capabilities and limitations . . ."

P9, Line 248: "Also from the spectra it can be seen, that for high optical depths of 6 and larger the response of the spectral radiance to a change in the cloud parameters strongly decreases, so results .. " –> "For high optical depths of 6 and above, the response of the spectral radiance to a change in the cloud parameters strongly decreases, so results. . ."

P9, Line 251: "The condensed water path can retrieved" –> "The condensed water path be can be retrieved"

P11, Line 312: "are retrieved" –> were retrieved

P11. Line 313: remove "measured using a FTIR spectrometer" and end the sentence at the comma.

P11, Line 317: "with different errors like biases . . ." –> with different errors such as biases . . .

P12, Line 320: "lead" –> led

P12, Line 322: Please mention how much the retrieved r_total differs from the true r_total.

P12, Line 325: Please put comma after "In those cases"

P12, Lines 344, 349: Please change "informations" to information

P12, Line 348: "can not" –> cannot

––––––––––––––––––––––––––––––––––

---

## Referee Comment (RC2) · Quentin Libois (Referee) · 19 Nov 2020

Review of « Retrieval of microphysical cloud parameters from EM-FTIR spectra measured in Arctic summer 2017», by Philipp Richter et al.

**General comments**

This study presents a retrieval algorithm for cloud physical properties (liquid and ice optical depths, liquid and ice effective radius) based on Fourier Transform InfraRed spectra in the range 770.9 – 1163.4 cm$^{-1}$, called TCWret. From these parameters, cloud water content, total effective radius and total optical depth are computed. The algorithm is based on radiative transfer simulations performed with LBLRTM and DISORT, coupled by LBLDIS. It also makes use of Mie calculations for single scattering properties of spherical particles, and of the database of Yang et al. (2013) for ice crystals single scattering properties. The algorithm is tested on synthetic spectra from Cox et al. (2016) that are perturbed in various ways to estimate the robustness of the retrieval. It is also applied on spectra measured on the Polarstern in the Arctic during summer 2017. These retrievals are compared to Cloudnet measurements acquired simultaneously on the Polarstern, that combine, lidar, radar and microwave radiometer measurements. The authors also demonstrate that the algorithm behaves as well as the previously published CLARRA algorithm, and allows to retrieve cloud optical thickness for thin clouds ($\tau < 6$), thus complementing other instruments that are more sensitive to optically thicker clouds. The retrieval of the cloud phase is not very reliable, hence the focus is on total condensed water. Likewise, the ice crystal habit is not retrieved by the algorithm. The algorithm also perfoms worse when a cloud is spread vertically compared to when it's geometrically thin, because of the assumption of geometrically thin cloud in the algorithm.

The manuscript is easy to read at first sight. However many details are missing when trying to better understand what the authors have done. Also, it is not particularly original in the sense that TCWret is very similar to existing algorithms. The sensitivity study is interesting but lacks of scientific perspectives. The observed spectra are not sufficiently explored to demonstrate the capability of such new instrument/algorithm. More generally the authors should stress more clearly why they developed a new algorithm while they refer to existing ones which seem to produce very similar results. Only with significant improvements on the method description, additional context and perspectives the paper might be worth being published in AMT.

**Specific comments**

1) Although the methodology is standard, many details are missing, which either points to deficiencies in the method, or prevents the reader from reproducing the algorithm. This is extensively detailed in the technical corrections but the most critical points are : poor treatment of ancillary data uncertainties in the retrieval, insufficient details about $S_a$ and $S_y$, limited usage of the matrix A (presented but not used further) and of the retrieval uncertainties.

2) To expand on the previous point, there is not sufficient discussion about the uncertainties in general. From the ancillary data used in the algorithm, to those on the retrieved parameters. Likewise, many figures are given for correlation coefficients, without any hint to their meaning or statistical significance. As a consequence, the reader does not really know what to expect from the numerous sensitivity tests, neither how to interpret the results. Statements like « are in agreement » do not help much.

3) The algorithm, which is the core of the study (given there is nearly no interpretation of the data acquired onboard Polarstern), is probably not sufficiently original, or different from existing algorithms (primarily CLARRA) to justify a dedicated paper. Unless the authors more strongly point why using CLARRA on their data would not have been possible or would have been worse than developing a dedicated tool.

4) More generally, the lack of physical interpretation of the results, in particular when it comes to real data, is frustrating. I understand that AMT may publish papers on pure methodology, but still some extended interpretation of the strengths and limits of the present algorithm would have been appreciated.

**Technical corrections**

title : the title suggests that the focus of the study is about the retrieval onboard Polarstern, while the paper is more about the algorithm validation. To be more consistent with the title, the paper should discuss in more details the retrievals performed in the Arctic. Conversely the title could be updated. Also, EM from "EM-FTIR" is never described.

l.2 : this « therefore » suggests that the reason for the study was the fact that FTIR is sensitive to clouds. I'd suggest to turn this into a more geophysical objective.

l.6 : CWP should not be italics I guess, neither "total" under "r"

l.7 : it is not clear here, neither in the text, why you specify that radiances under 600 cm$^{-1}$ are not used. Do you mean that usually such retrieval requires these longer wavelengths ? Does the FTIR acquire something outside the range 770.9 – 1163 cm$^{-1}$ defined earlier ? Please clarify

l.8 : is the validation the comparison to Clounet products or to synthetic testcases ? The structure of the abstract is not very clear in that sense

l.12: a correlation is not sufficient to describe a retrieval capability, because you could have a scaling factor in between. An RMSE or a relative error would be more relevant. Also it is not clear if these testcases are meaningful or irrealistic (no noise, very simplistic cloud), making these figures poorly informative.

l.14 : same comment for the correlation coefficient

l.16 : I believe the interest is for night conditions, otherwise powerful algorithms work well in the solar spectrum. If this is the case this should be emphasized in the abstract. More generally, the methodology should be put in a  broader context of instrumentation and inversion methods to demonstrate its added-value.

l.19 : radiation budget of what ? Surface, atmosphere, Earth system ?

l.20 : " with respect" sounds awkward

l.22 : radiative forcing is loosely defined (surface, TOA, upward or downward convention). Maybe say clearly " are assumed to cool the Earth"

l.24 : would it be useful to include ice clouds here instead of focusing on LWP ?

l.26 : LW flux is loosely defined. Do you mean broadband, or at any wavenumber ?

l.27 : please clarify where up- and downwelling fluxes are defined (below the cloud downward and at cloud top upward ?). Multiple scattering between what and what ? Inside cloud ? Between cloud and surface ? Also, the limits should be in terms of optical thickness, not in terms of LWP, no ? So maybe state that these LWP ranges are approximative and correspond to average clouds.

l.30 : larger → large ?

l.35 : quantify "high range". operate continuously

l.36: LWP < 15 g m$^{-2}$ ?

l.37 : maybe tell why MWR are only sensitive to liquid water. Maybe precise the frequency of such MWR.

l.38 : there exists a variety of instruments to study clouds, so maybe give more motivation to use FTIR instead of active instrumentation, VIS-NIR radiometers etc.

l.45 : maybe give a relevant reference for cloud studies with AERI. What do you mean by "built in particular" ? What's the difference with absorption FTIR ? If relevant, stress the importance of performing emission observations in your case.

l.46 : maybe specify both spectral resolutions

l.49 : is the algorithm related to the campaign, or to the instrument ? Could this algorithm be applied to other observations, potentially taken with another FTIR at a distinct resolution ?

l.54 : please give some insight into the differences between MIXCRA and CLARRA, here or later. Do both algorithms use far-infrared channels ? Also, if relevant, emphasize more clearly that you achieve similar results without far-infrared channels.

Figure 1 : greens dots

l.69 : FTIR acronym used earlier

l.71 : is this resolution before or after apodization ? What apodization is performed ? " so that a maximum..."

l.72 : "aperture". Not clear what you mean. There will always be contribution from the instrument itself. Maybe clarify that the field of view is not limited by the aperture but by the detector itself, if true. Maybe specify the field of view and focal length.

l.73 : "detector". What is MCT-detector ? Is there a window on the container to isolate the instrument but let the radiation get in ?

l.76 : the blackbodies. Please detail the absolute error of the instrument, or at least the temperature stability of the blackbodies.

l.77 : How long does it take to acquire an interferogram ?

l.84 : figures 6, 7, 8  do not have this unit ($^{-1}$ missing)

l.92 : detail what are aerosols and cloud properties. Only LWP or CWP from OCEANET? Do not mention LWP if ultimately you get vertical profiles of ice water content and liquid water content. Is effective radius retrieved for liquid and ice as well from Cloudnet approach ?

l.95 : acronym already defined. "inspired" is vague. Please review the main differences with those tools, to highlight that you're really developing something new. This is crucial for the relevance of this paper in AMT.

l.104 : does this mean that no uncertainty is attached to these gaseous profiles?

l.105 : "inhomogeneous". What does this mean ? Vertically inhomogeneous ?

l.107 : I guess DISORT does not handle effective radii, but single scattering properties (phase function and g). Can you clarify this.

l.109 : the role of LBLDIS is not clear

l.110 : please detail the characteristics of this distribution, and support the choice made with references or observations. Do you work with phase functions or simply asymmetry parameters ? Please clarify all these technical details.

l.112-116 : it is not clear what you take from LBLDIS and what you have computed on your own. Do the details on refractive indices refer to your work or to that of Turner (2014) ? I don't understand if you used the 1975 or 2005 database.

l.117 : ice water droplets is misleading for crystals that are practically not spherical…

l.118 : why not using the updated Warren (2008) database ?

l.119 : please detail how you computed SSPs from Yang et al. (2013) database. Using the same size distribution as for liquid clouds ? Also, please state here what choice is made in the inversion for ice crystal habit.

l.120-121 : not clear what "chosen … and modified" really means.

l.121 : it is not clear what offset, slope, and curvature are. Maybe provide a figure to illustrate this quantity on a given microwindow. Please explicit your observation vector. Does it comprise radiances, or these 3 parameters for each microwindow ?

l.123 : not clear. What do you mean regarding the trace gases ? That they are absent within the microwindow, or that they can still alter the retrieval in your case ?

l.124 : the information on cloud boundaries, as it does not appear in the state vector, should be provided beforehand. Also at which wavelength do the optical thicknesses correspond ?

l.131 : check the "n" indices, some are missing. Also $F(x_i)$ are radiances, while I would expect the 3 parameters defined above. This should be clarified if radiances or offset/slope/curvature are used.

l.132 : the $S_y$ matrix only includes observational errors, but no impact of the uncertainties on the ancillary information. How do you justify this ? What would be the impact of adding such uncertainties in $S_y$, as is commonly done in such retrieval techniques ? Also, how do you argue that $S_y$ is diagonal ? There is very likely correlation among channels of the instrument. That should be accounted for.

l.133 : how is $S_a$ built ? These details are needed.

l.140 : maybe precise how the averaging kernel matrix should be to be perfect, or how you can quantify the quality of the retrieval from this matrix. Is this matrix used further ? If not remove this information, although it can be informative if you add extra information on this.

l.150 : what is $D_n$ ? Diagonal unitary matrix ? Then why index n?

l.152 : what about the initial guess for $r_{eff}$ ? How many different optical depths ? What increment in this initial guess research ?

l.158 : what's the underlying assumption on extinction cross section of $Q_{ext}$ ? Is it justified ?

l.161 : not sure 3 decimals are necessary for a quantity that may vary with temperature

l.163 : "ice droplets of any shape" sounds awkward because droplets generally means spherical

l.170 : please detail why these total water quantities, which are built from quantities expected to be non reliable, should be more reliable.

l.178 : the covariance errors between retrieved state parameters is not discussed, while it is a very useful information

l.179 : please detail this error propagation method as it is not trivial, in particular because of the covariance errors.

l.179 : what residuum ? This alternative method is not clear.

l.180 : what residuum are used to compute $S_y$ ? It cannot be residuum at the end of the retrieval because this quantity is needed at the beginning of the iterations

l.183 : what about the radiative transfer code used to compute these spectra ? Is it consistent with your retrieval algorithm and the assumptions made on cloud particles SSPs ? What's the resolution of these spectra, how are they converted into your instrument resolution, assuming which spectral response function ?

l.188 : "inhomogeneous"

l.189 : does it mean that cases above are only pure ice clouds ?

l.191 : why setting spheres, which are for sure far from reality, and scatter way too much forward? Don't you think assuming another shape would be *a priori* more representative ?

l.193 : what's the rationale for this value of 0.2 radiance units ? How is this noise distributed across channels ? With any spectral correlation ?

l.194 : what temperature profile, the one assumed in the retrieval ? Is the actual one provided by Cox et al. (2016) ? More generally what information is provided by these latter authors along with the radiances. Why not recomputing the radiances on your own, because here we can't say what's the impact of having different codes for the radiance computation and the retrieval algorithm

l.195 : again, why such offset ? Is it based on experimental characteristics of the instrument/calibration unit ? Please give some insight into this value, maybe converting it into blackbody temperature offset.

l.198 : how do you observe such noise ?

l.200 : issue with section/subsection

l.204 : this information on cloud boundaries should have been provided earlier

l.205 : why not adding cloud top in the state vector, although the uncertainty would be large ? This would still be better than assuming a geometrically thin cloud

l.206 : again, droplets – spherical

l.207 : no information is provided about the uncertainty of the retrieved values, while this is critical here

l.210 : how are thin clouds selected ? What's the threshold ? Is there a flag when a retrieval does not work (for instance for optically thick clouds) ?

l.215 : it seems that there is absolutely no difference with CLARRA. Can you extend on this. Then, what's the point of developing a novel tool ? What differences would you expect ? What are the retrieved quantities for CLARRA ?

l.218 : what's the difference between CLARRA and TCWret for cloud position. In CLARRA it is in a layer (forced by an initial profile ?), but what about TCWret ? This is not clear.

l.220 : Do you mean that CLARRA was originally developed for an instrument covering these microwindows as well ? And you use those because they are available in the theoretical radiances ? Stress more strongly this difference in terms of far-infrared partial coverage.

l.223 : Why is there a difference between liquid water **path** and ice water **content** ? This is not clear. Do you compare only CWP or also optical depth and $r_{total}$ with Cloudnet ?

l.226 : this formula is useless, this is trivial

l.230 : how do you handle these errors when integrating ?

l.231 : where do these flags come from ? Can you explicit them to make it more understandable. Does it mean that you reject specific flags ?

l.234 : likewise, can you explain these flags. Why "No liquid water" in this subsection ?

l.236 : I don't understand. Where do you get LWC ? How can its integral be different from LWP ?

l.245 : when fitting this line, you always force it to pass through the origin ?

l.246 : this section is difficult to interpret, because we don't know to which extent the retrieval should be perfect, or what the residual errors come from. Inconsistencies between radiative codes or cloud assumptions, cloud position etc. The motivation for performing such unperturbed retrieval should be emphasized.

Figure 5 : would error bars on the retrieved value help interpret the figure ?

l.247 : why changing the τ limit from 8 to 6 ?

l.251 : can be retrieved

l.252 : it's very hard to comment on these correlation coefficients. What would you expect ?

l.255 : remind the reader what could be the inconsistencies there. I would actually expect something even worse if you assume spherical ice crystals

l.253 : « a closer view » is not very explicit

l.256 : it is not clear what the subset shown in Figure 9 is. What about the shape in Cox et al. (2016) database ? What's the difference between Figs. 9 and 5b?

l.262 : correlation coefficient r is not altered by the addition of noise ? Can you explain this ? In Fig. 6a it seems that optical depth exceeds 6 while you said (l.249) that you were excluding values larger than 6. Can you clarify.

l.266 : larger than what ? There was no underestimation so far.

l.276 : warning : a temperature offset won't have the same impact at all wavenumbers, so it's not equivalent

l.276 – 277 : consider rephrasing because this is not clear.

l.280 : at some point it would be helpful to understand how perturbations drive over- and underestimation of retrievals, providing some physical insight into how spectral radiance changes with the state parameters. Maybe showing the Jacobians at some point (earlier in the paper) would be useful.

l.283 : uniformly

l.292 : please clarify "different uncertainties in the spectral radiances"

l.300 : this is the first time error bars are shown (Fig. 11). Can you remind the reader where they come from. These should be discussed in more details throughout the paper

l.302 : the error bars are so large that points can hardly be out of this range

l.304 : I suspect in most cases the cloud extends over some height. Could you show the distribution of cloud altitudes or thickness according to Cloudnet as a complement ?

l.308 : I suspect assuming any other shape you've been working with would reduce the uncertainty due to ice crystal shape. Spheres might be the worst choice

l.310-315 : not sure this is useful because it only repeats previous parts of the paper. This is more generally the case for all this section, especially for such a short article. Consider removing This section 6.

l.318 : correlation ; retr**ie**ved

l.320 : leads

l.322 : again, explain why this criterion on $r_{liq}$ is working. What's the physical meaning ?

l.327 – 328 : already said

l.328 – 329 : this is a commonplace without further suggestions regarding the meaning of "careful"

l.329 : representation → assumption

l.330 : an error

l.339 – 340 : already said

l.350 : detail further why Cloudnet does not work well on thin clouds

l.352 : only one blackbody ?

---

## Referee Comment (RC3) · Anonymous Referee #3 · 20 Nov 2020

This paper presents an algorithm used to derive cloud properties (total optical depth, effective radius, and condensed liquid water) from a ground-based infrared interferometer. Results from the algorithm are compared against synthetic radiances, where the true cloud properties are known, assuming bias errors in the instrument / assumptions to evaluate the impact on the retrievals. It is then applied to downwelling radiance data observed during a multi-month summertime field campaign, with the results compared against a similar infrared retrieval algorithm and against an active+passive method (CloudNet). The results show high correlation between the retrieved properties using this new method, and those from the other methods.

My primary question is what is the new contribution of this work? The authors have stated that this algorithm is very similar to the MIXCRA and CLARRA algorithms, and indeed results from the latter are used to evaluate the results from this "new" algorithm. The authors need to state clearly how this algorithm differs, and ideally is an improvement, upon the previous two algorithms. As it is written, I do not think this paper merits being published until this question is answered.

The rest of my comments are much easier to address, and are offered to help the next version of the paper.

L22: your "in general" statement does a great disservice to a lot of previous research that has demonstrated that clouds have a cooling effect in tropical to mid-latitude locations, and a warming effect for many months in the polar regions (esp when the sun is below the horizon). This statement needs to be removed or totally rewritten

L 26: the downwelling LW radiative flux at the surface becomes less sensitive to changes in the LWP when the LWP > 40 g/m2 (by that Turner et al 2007 reference). But the LW radiative heating rate (and the SW radiative heating rate) profiles are sensitive to changes in LWP for LWP values that go much higher than this threshold (e.g., Turner et al. JAMC 2018)

L27: the sentence that starts "The blackbody-limit for. . ." is unclear – I don't understand the point you are trying to make

Multiple misspelled words throughout the document, e.g., on L33, L35, L53, L73, L187, L236, . . .. Please check entire document more carefully

Please add more references: e.g., to ARM on L44, AERI on L45, the other instruments on L90-L91

Is there a paper that describes your IR interferometer in detail? If not, then I believe that substantially more detail is needed here (but perhaps put in an appendix as to not disrupt the flow of the paper too much)

L73: this equation is a complex number, as the interferogram is almost certainly not symmetric. The Revercomb method shows that, if the instrument is well-behaved, the imaginary part is zero (with noise) and the radiance is the real part of this equation. This should be indicated here.

L79: This assumes that the blackbodies have an emissivity of 1.0. That is almost certainly not true, and even slight deviations of the emissivity from unity can affect the calibration. See Knuteson et al. JTECH 2004 part 2 on the AERI to see how they handle non-unity blackbody emissivity

L131: how do you determine Xa and Sa? This information is critical for understanding this retrieval.

L153: Why is the iterative method (i.e., starting with tau_total=tau_ice+tau_liq with tau_ice=tau_liq, getting a solution, and then individually retrieving tau_ice and tau_liq simultaneously) required if in the end you only really desire tau_total?

L157: Your algorithm retrieves four parameters, from which you derive three. Why not evaluate the 4 parameters you retrieve?

In same vein: your algorithm will provide uncertainties for these four parameters. Your paper would be stronger if you propagated those errors into your three derived parameters, and show the sensitivity of those uncertainties in tau_total, r_total, and CWP to the uncertainties of the four (tau_liq, tau_ice, r_liq, r_ice). This will come up again in L273 (see my question on that line below)

L205-208: where did these percentages (e.g., 98.50, 95.35, 98.28, etc) come from? For example, are you confident that of all of the clouds seen during the summer cruises that 98.50% of them had tau_total less than 6? If you only used your TCWret to determine the total cloud optical depth, then this is likely due to the sensitivity limit of the algorithm/data; does the CloudNet retrievals show the same fraction of cases with tau < 6?

L220: It seems that you have observations from your spectrometer in microwindows below 600 cm-1 (which is why we need more information on your instrument – see my comment above), but you chose not to use them in TCWret – Why not? Or is this paragraph only talking about the synthetic observations created by Cox that you used to evaluate your method? This paragraph is very unclear.

L273: does the fact that the CWP does not show the same dependence as tau_total imply that there are compensating errors (e.g., in r_total)?

L297: the instruments also have markedly different field-of-views

L306: If the assumed cloud temperature is too warm, then the retrieved optical depth will be too small – it is the only way to match the observed radiance. So the logic here seems to be backwards: using the cloud base height from the ceilometer would result in the cloud being too warm (relative to if the entire profile from a cloud radar was used to determine the cloud's temperature), and thus the TCWret's value should be LOWER than reality – but is not what is shown in Fig 11. This requires some more investigation / explanation, and will be important for a future submission (which I hope you will do).

---

## Referee Comment (RC4) · Anonymous Referee #4 · 30 Nov 2020

This paper presents an algorithm that allows to retrieve the properties of liquid clouds as well as ice clouds from high spectral resolution measurements in the mid-infrared range from the ground. The results of the algorithm are compared with synthetic measurements to test the performance of the algorithm. The algorithm is then applied to real observations from a measurement campaign that took place on a ship around Svalbard during the summer 2017. The results of the algorithm are compared with other algorithms that reproduce the same cloud properties for validation purposes.

First of all, I was very enthusiastic about the idea of evaluating this paper, because the subject is of undeniable interest to the community, especially because this type of

measurement is not often used to retrieve cloud properties (especially in the ice phase), but also because it could provide fundamental information on cloud microphysics and in particular the capabilities of current models to simulate a spectrally resolved measurement over a wide spectral range.

Generally speaking, the paper disappointed me both in the presentation of the results, where the lack of analysis, especially concerning the limits of the algorithm, is blatant, and in the scientific conclusions, which appear hasty and without concern for physical explanations. A number of passages may even be confusing and lead the reader to believe that the algorithm could realistically deal with ice cloud properties. I am thinking in particular of the part on the microphysics of ice clouds, and in particular the models of Yang et al. which, as the reader will discover much later, are not at all processed by the algorithm, which assumes a spherical shape for these particles.

The presentation of mathematical tools such as the averaging kernel, which seems to be very interesting for a posteriori analysis of the results and which are presented as such by the author, are absolutely not used. In short, the study looks confusing and not very informative. Moreover this study does not seem to bring anything new compared to other existing algorithms cited by the author or even used to validate the algorithm presented in this paper. The author absolutely does not allow the reader to get an idea of what the algorithm brings in comparison with CLARRA for example.

This study as presented does not hint the scientific level of the journal, the authors must moreover clearly demonstrate the novelty of their algorithm in relation to what is already done.

For all these reasons this study cannot be published in its present form.

Below are indicated in a more precise way all the remarks concerning the manuscript, which if followed could improve the paper.

Line 5: Could you give an explanation on why not using wavenumber smaller than 600

cm-1?

Line 27: In the sentence - The blackbody limit . . . - rephrase it, the sentence is not clear.

Line 32: performing

Line 50: the sentence is not well written you should change it with - because of low absorption by molecules the atmosphere is transparent and allows to see the signal emitted by clouds -.

Line 51: What do you mean by total effective radius? Is it the average or the sum? Please be more precise.

Line 98: - . . . new software. . .-, here it is important to tell the reader what is new in your algorithm compare to MIXCRA and more specifically compare to CLARRA.

Line 107: You really don't need to get this kind of technical details, moreover I am not sure about this explanation, the streams are used to integrate the scattering source function from scattering by any hemispherical direction in the direction of observation, you don't solve 16 different differential equation, please reformulate.

Line 111: could you please tell the reader which parameters are you talking on.

Line 114 to 116: The sentence - Because single . . . were used- is incomprehensible, please reformulate.

Line 117: -. . . they have large uncertainties . . .- what are you talking about? Indices, scattering parameters? Which parameters?

Line 118: My first thought was which shape are they using in their retrieval? But I understood later that you were not using these microphysical models, so why are you talking about it? Unless you want to specify here that you are not using it, but you may include these model in a futur algorithm, otherwise remove it because it is confusing we might think that your algorithm can undue such shape complexity.

Line 121: -... modified-, What is modified compare to Rowe et al.???

Equation 3: miss the n in Kn transpose.

Line 131: Could you specify how the Jacobian is computed in your algorithm?

Line 132: Could you tell a bit more on the variance of the measurement? How do you compute it, is it something furnish by the constructor of your instrument, please give some numbers!

Line 132: $S_a^{-1}$ is not an error but a variance-covariance matrix. How did you built it? Give some numbers.

140: Why the averaging kernel is an important quantity to characterize the retrieval quality?

Equation 8, 9, 10, 11: Please explain why you need an iterative procedure to compute the averaging kernel? Why not computing it once your algorithm converged without any iteration process beside the retrieval itself?

Line 153: How do you find the best tau_cw? Are you using LUT and find tau_cw from a simple least square method? Be more precise on this point!

Equation 13: This equation is not appropriate, you miss the particle number concentration N0, you should rewrite it by introducing a normalize size distribution and make explicitly appear N0, like: IWP=rho_ice*N0*Vmean*tau_ice/sig_ice

where sig_ice is the ice extinction (m-1) Vmean is the mean volume computed over the normalized size distribution...

Line 162: V0 is not the volume of an ice droplet but the mean volume computed over the normalized size distribution. Same for ext, it should be the extinction coefficient of ice (m^-1)...

Line 164: What is - cVo - ? Same for Cext? Please define all the variables you are

using in your equation!

Line 180: This sentence is very confusing - Here the maximum ... - how can you use the residual to define Sy. The residual will be known at the end of the iteration process and needs the Sy matrix in the iteration process! Sy should be defined before and should be provided posterior to the retrieval, by the instrument manufactor, which provide NeSR and absolute accuracy of the instrument!

Line 191: Again why not using the Non-spherical model here? Or at least use it to define a forward model error due to the simplest assumption of spherical ice particles.

Line 193: How did you choose this value of 0.2 (mW/str/m2/cm-1)? Please give some justification.

Line 194: How did you disturb the Temperature, is it like a white noise, do you use any gaussian statistics?

Line 195: What do you mean by a radiance offset, is it like a bias? If so why did you choose this value, is it related to the instrument you are using? If yes give some justification/reference!

Line 202,203: Please describe this instrument (Vaisala CL51), and tell the reader why you can take the cloud top height (CTH) from this instrument and give some accuracy to this value.

Line 206: What are the errors introduce by the assumption of spherical ice droplets?

Section 4.3: Why do you present the results here when there is a specific section on the subject afterwards (section 5)?

Section 4.4: It seems from this section that your algorithm is identical to the CLARRA one. You really need to tell the reader the main difference and how they can complement each other!

Line 235: what do you mean by - Incorporated - ?

Line 239: The sentence is not clear, please rephrase it!

Line 248: - tau_cw > 6 are ... available = It is not clear what you want to say here! Rephrase it.

Line 248: - Also the spectra ... - From which spectra are you talking about? This something that you say but don't show... A figure would has help to proof it.

Line 250: - The condensed water path can retrieved - strange! - can be retrieved - I guess - without the exact knowledge of ice ... - What do you mean? Is it always the case? Or only when the ice fraction is low compare to the liquid one? Please be more precise!

Line 249: This value of 20 microns is very subjective, how can we state that this value is an indicator of the accuracy of the retrieval!! you have presented some mathematical tools ,like the Averaging kernel or a-posteriori errors that can be used as an indicator for bad or good retrievals, why not using them?

Line 258: - However ...- Why did you present the Ping Yang model then? One could think that this retrieval algorithm might include some non-spherical particles which would have been very interesting, because it is well known that ice particle are non-spherical. So I am wondering what this algorithm is bringing to the community, The method (optimal estimation) use to make these restitutions is sophisticated enough to integrate the errors of the model in order to propagate them on the retrieved parameters, from the definition of the forward model variance-covariance matrix. Here the author could have used this formalism to take into account the error linked to the assumption of using spherical particles for ice for exemple ...

Line 264: What is this factor (slope or correlation)?

Section 5.2: Why The posteriori errors attached to each parameter are not given? It would have been interesting to add them in order to be able to evaluate the accuracy of each algorithm.

Line 302: This sentence is not clear, the points are always presented in the center of the error bar with a length equal to a factor times the standard deviation, therefore the points can't be outside of the standard deviation. . .

Line 302 to 307: This part is not clear we don't understand what you are trying to explain. . .

Line 327.328,329: Instead of saying this kind of evidence, it would have been interesting to give some information on the accuracy that the instrument should have reach in order to get accurate enough retrieved information of cloud.

Line 338: If cloudnet is the reference you should have said the results from TCWret show some overestimation of . . .

Line 339, 340: - However, retrieval of microphysical cloud parameters . . . - one could have wait something a bit more precise, which parameter is well retrieved ans which one not, why not using the averaging-kernel to give some indication. The a posteriori error analysis is completely non-existent, it could have highlighted the limitations of the algorithm by correlating, for example, to the thermodynamic conditions or to the cloud column (presence of several cloud layers, high humidity, cloud layer close to the ground, fractional coverage).... which strongly not serves this study.

Section 7: This section is not necessary and does not add anything to the paper, the conclusion is already in the previous section.